

# The regional EUROpean atmospheric transport inversion COMparison, EUROCOM: first results on European wide terrestrial carbon fluxes for the period 2006-2015

Guillaume Monteil[1], Grégoire Broquet[2], Marko Scholze[1], Matthew Lang[2], Ute Karstens[3], Christoph Gerbig[4], Frank-Thomas Koch[8,4], Naomi E. Smith[6], Rona L. Thompson[7], Ingrid T. van der Laan-Luijkx[6], Emily White[5], Antoon Meesters[9], Philippe Ciais[2], Anita L. Ganesan[5], Alistair Manning[5], Michael Mischurow[1], Wouter Peters[10], Philippe Peylin[2], Jerôme Tarniewicz[2], Matt Rigby[5], Christian Rödenbeck[4], Alex Vermeulen[3], and Evie M. Walton[5]

[1]Dep. of Physical Geography and Ecosystem Science, Lund University, Sweden
[2]Laboratoire des Sciences du Climat et de l'Environnement, LSCE/IPSL, CEA-CNRS-UVSQ, Université Paris-Saclay, Gif-sur-Yvette, France
[3]ICOS Carbon Portal, Lund University, Sweden
[4]Max Planck Institute for Biogeochemistry, Jena, Germany
[5]University of Bristol, Bristol, United Kingdom
[6]Meteorology and Air Quality, Wageningen University and Research, Wageningen, the Netherlands
[7]NILU – Norsk Institutt for Luftforskning, Kjeller, Norway
[8]Deutscher Wetterdienst, Germany
[9]Vrije Universiteit Amsterdam, The Netherlands
[10]Centre for Isotope Research, University of Groningen, Groningen, the Netherlands

**Correspondence:** G. Monteil (guillaume.monteil@nateko.lu.se)

**Abstract.** Atmospheric inversions have been used for the past two decades to derive large scale constraints on the sources and sinks of $CO_2$ into the atmosphere. The development of high density in-situ surface observation networks, such as ICOS in Europe, enables in theory inversions at a resolution close to the country scale in Europe. This has led to the development of many regional inversion systems capable of assimilating these high-resolution data, in Europe and elsewhere. The EURO-
COM project (EUROpean atmospheric transport inversion COMparison) is a collaboration between seven European research institutes, which aims at producing a collective assessment of the net carbon flux between the terrestrial ecosystems and the atmosphere in Europe for the period 2006-2015. It aims in particular at investigating the capacity of the inversions to deliver consistent flux estimates from the country scale up to the continental scale.

The project participants were provided with a common database of in-situ observed $CO_2$ concentrations (including the
observation sites that are now part of the ICOS network), and were tasked with providing their best estimate of the net terrestrial carbon flux for that period, and for a large domain covering the entire European Union. The inversion systems differ by the transport model, the inversion approach and the choice of observation and prior constraints, enabling us to widely explore the space of uncertainties.

This paper describes the intercomparison protocol and the participating systems, and it presents the first results from a ref-
erence set of inversions, at the continental scale and in four large regions. At the continental scale, the regional inversions





support the assumption that European ecosystems are a relatively small sink (-0.21±0.2 PgC/year). We find that the convergence of the regional inversions at this scale is not better than that obtained in state-of-the-art global inversions. However, more robust results are obtained for sub-regions within Europe, and in these areas with dense observational coverage, the objective of delivering robust country scale flux estimates appears achievable in the near future.

## 1 Introduction

The carbon budget of Europe has been explored in several large scale synthesis studies, such as the CarboEurope-Integrated Project (Schulze et al., 2009) and the REgional Carbon Cycle Assessment and Processes project (RECCAP; Luyssaert et al., 2012), to name a few. Although these have helped refining the knowledge of the European carbon cycle, large uncertainties remain regarding the quantification of the flux between terrestrial ecosystems and the atmosphere, usually quantified as the Net Ecosystem Exchange (NEE), i.e. the sum of emissions (TER, i.e. autotrophic and heterotrophic respiration) and uptake (GPP, i.e. photosynthesis) of carbon by ecosystems to and from the atmosphere, or alternatively NBP, which includes the impact of ecosystem disturbances (fires, land use change, etc.). For instance, Luyssaert et al. (2012) report average estimates of European land carbon sink in a -200 to -360 TgC/year range for the years 2001-2005, depending on the estimation method, and each of these estimates are provided with large uncertainties (with 1-sigma relative uncertainties of 50 to 100%). Confronting the ensemble of results from different syntheses, Reuter et al. (2017) report annual land-atmosphere flux ranging from -400±420 TgC/year up to -1030±470 TgC/year in the 2000s. Beyond the annual long-term budget, the year to year annual flux variations are also poorly known (Bastos et al., 2016). In practice, the lack of a robust and precise quantification of the natural $CO_2$ fluxes in Europe limits our ability to understand the links between the NEE flux and external forcings such as e.g. meteorological variability (including the impact of extreme events like droughts and cold spells) and trends (Ciais et al., 2005; Maignan et al., 2008) or land use change (Naudts et al., 2016), and to forecast the evolution of the land sink in Europe, in the context of global climate change.

Despite the large uncertainties, there is a growing demand from the policy makers and the society in general for more accurate and relevant numbers, such as estimates of the national budgets of $CO_2$ fluxes, these demands being reinforced by the Paris Agreement. For instance, the European Commission (under the VERIFY and CHE H2020 projects) is supporting the development of observation based monitoring systems for estimating $CO_2$ fluxes at national to sub-national scales, with a clear interest in both land ecosystems fluxes and the anthropogenic emissions.

Atmospheric transport inversions rely on transport models and statistical methodologies to derive the most likely estimates of $CO_2$ fluxes given large datasets of observed atmospheric $CO_2$ concentrations and a prior information provided in general by ecosystem models. Global inversion systems, using coarse resolution global transport models (typically >2°), have so far been the dominant tool for producing top-down estimates of NEE fluxes. The coordination of the inverse modelling community through intercomparison exercises with ≈10 global inverse modelling systems, such as that conducted in the frame of the TRANSCOM and RECCAP projects (Law et al., 1996; Gurney et al., 2002; Patra et al., 2008; Peylin et al., 2013) have been valuable for understanding the strengths and weaknesses of global inversions and to characterise the real uncertainty of the





different estimates. However, despite this long term effort, global inversions remain limited by the coarse resolution of the
50  transport models they rely on, as these do not allow a proper representation of observation sites in regions with complex
orography or nearby large anthropogenic $CO_2$ emissions and do not reproduce the high-resolution spatial variability of the
$CO_2$ concentrations that is captured by dense networks.

Regional scale inversions started to emerge about a decade ago. They rely on mesoscale transport models (at 1° down to
10 km resolution), capable of better representing the spatial and temporal variability of concentrations observed by dense
networks of $CO_2$ observations, such as that of the Integrated Carbon Observation System (ICOS) in Europe. In particular the
models should be able to account for $CO_2$ fluxes at a scale that does not smoothes too much the hot spots of fossil fuel $CO_2$
emissions in cities and industrial areas. They demonstrated some potential to solve for continental to subcontinental budgets at
the monthly scale (e.g. Peters et al. (2007); Rödenbeck et al. (2009); Schuh et al. (2010); Gourdji et al. (2012); Broquet et al.
(2013); Meesters et al. (2012)). However, the sparse efforts for routinely producing regional inversion estimates (beyond the
scope of specific studies), as well as (up until recently) the difficult access to long-term time series of quality-controlled $CO_2$
data from many sites in Europe, were limiting their development. For those reasons, most synthesis studies up to now kept
relying on results from global scale inversions for European NEE, based on networks of global background sites.

The ICOS atmospheric network (icos-atc.lsce.ipsl.fr) is now operational and its number of stations should regularly increase
from the current 19 labelled stations towards at least 34 stations, run by currently twelve and hopefully in the future more
European member states. Precursor networks such as those set-up in the framework of the CarboEurope and GHG-Europe
projects (Ramonet et al., 2010) and the ICOS preparatory phase provide a robust basis for regional inversions during the pre-
ICOS decade. The ICOS Carbon Portal (www.icos-cp.eu) has been set-up to support the exchange of observational data and
elaborated products related to the carbon cycle, such as $CO_2$ fossil fuel flux maps. In addition to in-situ data, the development
of satellite observations of $CO_2$ following the launch of GOSAT (Kuze et al., 2009) in 2009 and OCO-2 in 2014 (Crisp et al.,
2004) should further densify the observation coverage, in particular with the foreseen European constellation of $CO_2$ high
resolution imagers of the Copernicus Anthropogenic $CO_2$ Monitoring mission (CO2M; Pinty et al., 2017), starting from 2025.
The use of mesoscale transport model will then become necessary to fully exploit the potential of these large datasets with
observations at high spatial and temporal resolution.

In this context, the EUROCOM (EUROpean atmospheric transport inversion COMparison) project aims to coordinate a Eu-
ropean effort to improve the knowledge on the NEE based on an ensemble of long-term European scale inversions (i.e. covering
geographical Europe). The participating groups were tasked with performing an ensemble of mesoscale $CO_2$ inversions of the
European NEE for the period 2006-2015, following a protocol described further in this document. A large dataset of surface
$CO_2$ observations, combining measurements from several European networks and individual research stations was compiled
and provided to the participants. A total of seven research groups participated in the project, producing an ensemble of more
than (to date) 10 inversions (including sensitivity experiments). The EUROCOM project is therefore one of the first regional
inversion intercomparisons, and the first one at such a scale dedicated to the European NEE.

This paper presents the protocol of the intercomparison and a first set of analyses of the results. The inversions were provided
by six different groups, with six inversion systems: PYVAR-CHIMERE (Broquet et al., 2011; Fortems-Cheiney et al., 2019,





developed at LSCE, France); LUMIA (Lund University Modular Inversion Algorithm) (Monteil and Scholze, 2019), developed
at Lund University (Sweden) as part of the EUROCOM project; CarboScope-Regional (Kountouris et al., 2018a, b, developed
at MPI-Jena, Germany); FLEXINVERT+ (Thompson and Stohl, 2014, from NILU, Norway)); NAME-HB (White et al., 2019,
from the University of Bristol, United Kingdom) and CarbonTracker Europe (Peters et al., 2010; van der Laan-Luijkx et al.,
2017), from the University of Wageningen, the Netherlands.

The analysis focuses on assessing whether these regional inversions help to better characterise the annual to monthly budgets
of NEE for the whole Europe. It also provides first insights on the robustness of the sub-continental flux estimates. The
advantages and current limitations of regional inversions, compared to global ones, are also discussed. Forthcoming studies
will provide a more in-depth analysis of the whole inversion ensemble, with the aim to better understand the strengths and
weaknesses of the regional inversion, characterise their sources of uncertainties and attempt at supporting the improvement of
both the regional inversion techniques and the design of the European observation network.

The manuscript is organised in five sections. Section 2 briefly summarises the theoretical background behind atmospheric
transport inversions. Section 3 details the inversion protocol, the participating inverse modelling systems, and the input products
(fluxes and observations) shared within EUROCOM for conducting the inversions. Results are presented in Section 4 and then
discussed in Section 5. Finally, Section 6 summarises the paper and provides some remarks on the future of the EUROCOM
collaboration, and on regional inverse modelling in general.

## 2 Inverse modelling methodology / terminology

The theoretical framework of the atmospheric inverse transport modelling has been extensively detailed in past publications
(e.g. Enting, 2002; Rayner et al., 2018). Here we only give a brief overview of the basic principles, to facilitate the compre-
hension of the paper for readers unfamiliar with the approach and to remind of some of the components discussed in detail in
Section 3.

Bayesian atmospheric inversions rely on the fact that observed spatio-temporal gradients of $CO_2$ in the atmosphere reflect the
distribution of carbon exchanges between the atmosphere and other carbon reservoirs. The link between the net $CO_2$ exchange
at the surface and the $CO_2$ concentrations in the atmosphere is established by a forward atmospheric transport model. A
first set of modelled $CO_2$ concentrations ($\mathbf{y^m} = H(\mathbf{x})$) is computed at the time and location of real observations ($\mathbf{y^o}$), based
on a prior assumption of what the $CO_2$ fluxes are ($\mathbf{x_b}$). The mismatch between the modelled and observed concentrations
($\delta\mathbf{y} = H(\mathbf{x}) - \mathbf{y^o}$) is used to derive a correction $\delta\mathbf{x}$ to the prior flux estimate $\mathbf{x_b}$. The posterior flux estimate ($\mathbf{x} = \mathbf{x_b} + \delta\mathbf{x}$) then
represent the best statistical compromise between fitting the observations and limiting the departures to the prior, accounting
for the statistical distribution of uncertainties in both observations and prior fluxes.

The vector $\mathbf{x}$ is called the control vector. It contains all the parameters that the inversion can adjust. In our case it contains
at least the terrestrial ecosystem component of the $CO_2$ fluxes. It can also contain other adjusted parameters such as bias
or boundary concentration terms. The operator $H$, which establishes the deterministic relationship between a given control
vector $\mathbf{x}$ and the corresponding modelled concentrations $\mathbf{y^m}$ is called the observation operator. It encompasses the transport



model, but also the impact on the modelled concentrations of any input of the transport model that is not further adjusted in the inversions (prescribed anthropogenic emissions, boundary conditions, etc.).

Following the Bayesian approach and using classical Gaussian errors hypothesis the problem reduces to the find the posterior

control vector $\mathbf{x_a}$ that minimises the cost function $J(\mathbf{x})$, defined as:

$$J(\mathbf{x}) = \underbrace{\frac{1}{2}\delta\mathbf{x^T}\mathbf{B}^{-1}\delta\mathbf{x}}_{J_b} + \underbrace{\frac{1}{2}\delta\mathbf{y^T}\mathbf{R}^{-1}\delta\mathbf{y}}_{J_{obs}} \tag{1}$$

The prior error covariance matrix $\mathbf{B}$ contains a representation of the uncertainties on the prior control vector $\mathbf{x_b}$ and the error covariance matrix $\mathbf{R}$ contains an estimation of the uncertainties in the model data mismatches $\delta\mathbf{y}$ (observational uncertainty, and uncertainties from the observation operator such as model error, representation error and aggregation error). Departing

from the prior control vector $\mathbf{x_b}$ increases $J_b$, and improving the fit to the observations reduces $J_{obs}$. $\mathbf{B}$ and $\mathbf{R}$ modulate the relative weight of each departure to the prior and to the observations in $J$.

The exact specifications of $\mathbf{B}$ and $\mathbf{R}$ affect to a certain extent the outcome of an inversion. For practical reasons, the error covariance matrix for the observations, $\mathbf{R}$ is usually defined as a diagonal matrix with the measurement and model uncertainty ($\sigma$) for each observation site specified on the diagonal. Potential error correlation between observations are typically dealt

with by limiting the density of observations or inflating their individual uncertainties. The diagonal elements of the prior error covariance matrix $\mathbf{B}$ contains the uncertainties on the prior control parameters (typically here the NEE at the grid scale). The off-diagonal elements, corresponding to the covariances between uncertainties in different control parameters, are difficult to specify because the uncertainties in the NEE estimates have hardly been characterised and quantified (Kountouris et al., 2015). They are however a critical component of the inversion as they determine how independently from each other the different

components of the control vector can be adjusted. The inversions in this study follow different implementations of this general methodology, listed in Section 3.3.2.

The optimal control vector $\mathbf{x_a}$ can be solved for using different solution methods. Here we only briefly recall the methods employed by the systems in this study (variational and sequential ensemble approaches, and Markov Chain Monte Carlo), more information on these methods is given in Rayner et al. (2018) and references therein.

The variational method minimises $J(\mathbf{x})$ based on iterative gradient descent methods. Efficient implementations of this method rely either on the availability of the adjoint of the transport model or pre-computed transport Jacobian matrices representing the sensitivity of the observation vector to the control vector. The Monte Carlo approach directly samples the cost function, and in the case of the Markov chain Monte Carlo (MCMC) approach, the samples form a Markov chain, i.e. each sample is not obtained independently, but rather a perturbation of the last previously accepted sample. This allows non-Gaussian PDFs

to be used in the inversion, and allows the specification of uncertainties to be explored in so-called "hierarchical" Bayesian frameworks (Ganesan et al., 2014; Lunt et al., 2016). Finally, the Ensemble Kalman Filtering (EnKF) directly derives $\mathbf{x^a}$ following its analytical formulation based on the reduction of the dimensions of the problem through the split of the inversion into sequential windows, and based on on the computation of the matrices involved in the EnKF formulation through an ensemble Monte Carlo approach.


## 3   Protocol and participating models

The main product requested from the participating groups was a monthly gridded estimate of the net land-atmosphere $CO_2$ exchange (Net Ecosystem Exchange, NEE) over the period 2006 to 2015, covering at least the area 15°W-35°E by 33°N-73°N, at a 0.5° by 0.5° spatial resolution (independently of the actual resolution of the inversions).

The participants were to a certain extent free to choose their "best" inversion set-up except for a few restrictions and guidelines set out in the EUROCOM inversion intercomparison protocol. The only mandatory requirement for all inversion system was to use a common dataset of anthropogenic $CO_2$ emissions (fossil fuel combustion, cement production and large-scale fires) as detailed below (Section 3.2.2) and to use only atmospheric observations from a common dataset, prepared specifically for the EUROCOM project (Section 3.1). The precise data selection within that database (selection of observation sites, and selection of observations at each site) and the definition of observation uncertainties were also left to the modellers.

The treatment of boundary conditions, of meteorological input data, the use of an ocean flux and the precise specification of uncertainties (on the prior and on the observations) were left to the modellers. A set of fluxes (prior NEE, anthropogenic and ocean fluxes) was made available to the modellers through a data repository hosted at the ICOS Carbon Portal, along with the common observation database.

Note that we use the term NEE (sum of photosynthesis and ecosystem respiration) for the posterior fluxes throughout the paper, because this is what the priori flux estimates from the terrestrial ecosystem models represent. However, strictly speaking, the inversions optimise the flux that is not explained by the prescribed anthropogenic (and ocean) fluxes. This includes the effect of ecosystem disturbances (land use, land management, biotic effects) but also any errors in the prescribed fluxes.

### 3.1   Common atmospheric observation database

A comprehensive data set of atmospheric $CO_2$ concentration observations in Europe was compiled as input for the inversion systems, on the basis of the GLOBALVIEWplus v3.2 Observation Package (ObsPack), a product compiled and coordinated at NOAA's Earth System Research Lab together with the ICOS Carbon Portal (Cooperative Global Atmospheric Data Integration Project, 2017). The data set was further extended by including measurements that had been collected in several national and EU-funded projects, like CarboEurope-IP, GHG-Europe, and during the preparatory phase of the Integrated Carbon Observation System (ICOS) Research Infrastructure. Finally, for two stations, the data were obtained from the World Data Center for Greenhouse Gases (https://gaw.kishou.go.jp/).

Compared to the original GLOBALVIEWplus product, we added time series from nine measurement stations and partly complemented time series at two stations. The data sets were harmonised with respect to format and sampling interval, and provided in the ObsPack format (Masarie et al., 2014). The original datasets and data providers of the time series are reported in Table 1, and the locations of the observation sites are also shown in Figure 1.

The majority of sites (35 out of 39) sample concentrations continuously (i.e. hourly or more frequent); 18 sites are tall towers (intake height > 50m), some with observations available at different levels, in which case only the upper level was used (as more difficult for the transport models to represent concentration gradients close to the ground).



The modellers were free to refine the observation selection according to the the ability of their inversion systems to simulate specific stations, and in particular to use their preferred approach to select data within a day (i.e. use of all the observations within a time frame or use of an average of the observations, etc.). The precise observation selection approaches are discussed further in Section 3.3.3, and a full full comparison of the observation assimilated by each system is provided in Figures SI1 and SI2.


**Figure 1.** EUROCOM domain (pale blue grid with the 0.5° resolution) and location of the observation sites. The size of the dots is proportional to the number of months with at least one observation available (in the common observation database, not all observations are used in the inversions), and the colour map shows the altitude of the sites (height above ground + sampling height). The four regions used in the analysis are also represented: Western Europe (green), Southern Europe (blue), Central Europe (yellow) and Northern Europe (grey).





## 3.2 Prior and prescribed CO$_2$ fluxes

All groups split the total surface–atmosphere CO$_2$ flux in three or four categories: biosphere (NEE, optimised), ocean (sea-
atmosphere CO$_2$ exchanges, prescribed or optimised), anthropogenic (prescribed) and biomass burning (prescribed, used by
LUMIA and FLEXINVERT+).

### 3.2.1 Terrestrial-ecosystem fluxes

Atmospheric inversions usually rely on NEE simulations from terrestrial ecosystem models to provide the prior value of the
NEE component of the control vector (as defined above in Section 2). Within EUROCOM four different simulations of gross
(GPP and ecosystem respiration) and net (NEE) terrestrial biosphere fluxes: three from process-based models (ORCHIDEE,
LPJ-GUESS and SiBCASA), and one from a diagnostic model (VPRM). Two of the four models (ORCHIDEE and LPJ-
GUESS) are providing input for the Global Carbon Project annual global CO$_2$ assessment (Le Quéré et al., 2018).

- **ORCHIDEE (used by PYVAR-CHIMERE, FLEXINVERT+ and NAME-HB) :** ORCHIDEE (Krinner et al., 2005))
  computes carbon, water and energy fluxes between the land surface and the atmosphere and within the soil-plant con-
tinuum. The model computes the Gross Primary Productivity with the assimilation of carbon based on Farquhar et al.
  (1980) for C3 plants. Land cover changes (including deforestation, regrowth and cropland dynamic) were prescribed
  using annual land cover maps derived from the harmonised land use data set (Hurtt et al., 2011) combined with the
  ESA-CCI land cover products.

- **LPJ-GUESS (used by LUMIA):** LPJ-GUESS (Smith et al., 2014) combines process-based descriptions of terrestrial
ecosystem structure (vegetation composition, biomass and height) and function (energy absorption, carbon and nitrogen
  cycling). Vegetation is dynamically simulated as a series of replicate patches, in which individuals of each simulated plant
  functional type (or species) compete for the available resources of light and water, as prescribed by the climate data. LPJ-
  GUESS includes an interactive nitrogen cycle. The simulation used here is forced using the WFDEI meteorological data
  set (Weedon et al., 2014) and produces 3-hourly output of gross and net carbon fluxes.

- **SiBCASA (used by CTE):** SiBCASA (Schaefer et al., 2008) combines the parameterisation of the Simple Biosphere
  model (SiB) with the biogeochemistry of the Carnegie-Ames Stanford Approach (CASA) calculating the exchange of
  water, carbon and energy between 25 soil layers, plants, and the atmosphere. The rate of photosynthesis is found using
  the Ball-Berry-Woodrow model of stomatal conductance (Ball et al., 1987), and C3 and C4 vegetation types are treated
  separately in the kinetic enzyme model of Farquhar et al. (1980). The simulation used here is forced using meteorological
inputs from ERA-Interim, and run it on a 10 minute time step and a spatial resolution of 1x1 degrees.

- **VPRM (used by CarboScope-Regional)**: VPRM (Mahadevan et al., 2008) calculates photosynthetic uptake based on
  a light-use efficiency approach and temperature dependent ecosystem respiration. It uses ECMWF operational meteoro-
  logical data for radiation and temperature, the SYNMAP land cover classification (Jung et al., 2006), as well as MODIS



| Code | Station Name | Lat (°N) | Lon (°E) | Alt (m.a.s.l.) | Intake (m.a.g.l.) | C/F | Period | Dataset | Data Provider |
|---|---|---|---|---|---|---|---|---|---|
| BAL | Baltic Sea | 55.350 | 17.220 | 3 | 25 | F | 2006-2011 | GV+ v3.2 | NOAA |
| BIK | Bialystok | 53.232 | 23.027 | 183 | 300 | C | 2006-2007 | preICOS | MPI-BGC |
| BIR | Birkenes | 58.389 | 8.252 | 219 | 2 | C | 2015 | GV+ v3.2 | NILU |
| BRM | Beromunster | 47.190 | 8.175 | 797 | 212 | C | 2012-2015 | GV+ v3.2 | Uni.Bern |
| BSC | Black Sea Coast | 44.178 | 28.665 | 0 | 5 | F | 2006-2011 | GV+ v3.2 | NOAA |
| CES | Cabauw | 51.971 | 4.927 | -1 | 200 | C | 2006-2015 | GV+ v3.2 | ECN |
| CIB | Centro de Investigacion de la Baja Atmosfere | 41.810 | -4.930 | 845 | 5 | F | 2009-2015 | GV+ v3.2 | NOAA |
| CMN | Monte Cimone | 44.180 | 10.700 | 2165 | 12 | C | 2006-2015 | WDCGG | IAFMCC |
| CRP | Carnsore Point | 52.180 | -6.370 | 9 | 14 | C | 2010-2013 | preICOS | EPA |
| ELL | Estany Llong | 42.575 | 0.955 | 2002 | 3 | F | 2008-2015 | GV+ v3.2 | ICTA-ICP |
| GIF | Gif sur Yvette | 48.710 | 2.148 | 160 | 7 | C | 2006-2009 | preICOS | LSCE |
| HEI | Heidelberg | 49.417 | 8.674 | 116 | 30 | C | 2006-2015 | GV+ v3.2 | UHEI |
| HPB | Hohenpeissenberg | 47.801 | 11.024 | 985 | 5 | F | 2006-2015 | GV+ v3.2 | NOAA |
| HPB | Hohenpeissenberg | 47.801 | 11.010 | 934 | 131 | C | 2015 | GV+ v3.2 | DWD-HPB |
| HTM | Hyltemossa | 56.098 | 13.419 | 115 | 150 | C | 2015 | GV+ v3.2 | Uni.Lund-CEC |
| HUN | Hegyhátsál | 46.950 | 16.650 | 248 | 115 | C | 2006-2015 | GV+ v3.2 | HMS |
| JFJ | Jungfraujoch | 46.550 | 7.987 | 3570 | 10 | C | 2006-2015 | GV+ v3.2 | KUP |
| JFJ | Jungfraujoch | 46.550 | 7.987 | 3570 | 10 | C | 2010-2015 | GV+ v3.2 | EMPA |
| KAS | Kasprowy | 49.232 | 19.982 | 1989 | 5 | C | 2006-2015 | GV+ v3.2 | AGH |
| LMP | Lampedusa | 35.510 | 12.610 | 45 | 5 | F | 2006-2015 | GV+ v3.2 | NOAA |
| LMP | Lampedusa | 35.520 | 12.620 | 45 | 8 | C | 2006-2012 | preICOS | ENEA |
| LMU | La Muela | 41.594 | -1.100 | 571 | 79 | C | 2006-2009 | preICOS | ICTA-ICP |
| LUT | Lutjewad | 53.404 | 6.353 | 1 | 60 | C | 2006-2015 | GV+ v3.2 | Uni.Groningen |
| MHD | Mace Head | 53.326 | -9.904 | 5 | 15 | C | 2006-2015 | GV+ v3.2 | LSCE |
| NOR | Norunda | 60.086 | 17.479 | 46 | 101 | C | 2015 | GV+ v3.2 | Uni.Lund-CEC |
| OPE | Observatoire Pérenne de l'Environnement | 48.562 | 5.504 | 390 | 120 | C | 2011-2015 | preICOS | LSCE |
| OXK | Ochsenkopf | 50.030 | 11.808 | 1022 | 163 | F | 2006-2015 | GV+ v3.2 | NOAA |
| OXK | Ochsenkopf | 50.030 | 11.808 | 1022 | 163 | C | 2006-2007 | preICOS | MPI-BGC |
| PAL | Pallas | 67.973 | 24.116 | 565 | 5 | C | 2006-2015 | GV+ v3.2 | FMI |
| PRS | Plateau Rosa | 45.930 | 7.700 | 3480 | 10 | C | 2006-2015 | GV+ v3.2 | RSE |
| PUI | Pujio | 62.910 | 27.655 | 232 | 84 | C | 2011-2014 | preICOS | FMI |
| PUY | Puy de Dôme | 45.772 | 2.966 | 1465 | 15 | C | 2006-2015 | GV+ v3.2 | LSCE |
| RGL | Ridge Hill | 51.998 | -2.540 | 204 | 90 | C | 2012-2015 | GV+ v3.2 | Uni.Bristol |
| SMR | Smear/Hyytiala | 61.847 | 24.295 | 181 | 125 | C | 2012-2015 | GV+ v3.2 | UHELS |
| SSC | Sierra de Segura | 38.303 | -2.590 | 1349 | 20 | C | 2014-2015 | GV+ v3.2 | ICTA-ICP |
| SSL | Schauinsland | 47.920 | 7.920 | 1205 | 12 | C | 2006-2015 | GV+ v3.2 | UBA |
| STM | Station M | 66.000 | 2.000 | 0 | 7 | F | 2006-2015 | GV+ v3.2 | NOAA |
| TAC | Tacolneston | 52.518 | 1.139 | 56 | 185 | C | 2013-2015 | GV+ v3.2 | Uni.Bristol |
| TRN | Trainou | 47.965 | 2.112 | 131 | 180 | C | 2006-2015 | preICOS | LSCE |
| TTA | Angus tall tower | 56.555 | -2.986 | 400 | 222 | C | 2013-2015 | GV+ v3.2 | Uni.Bristol |
| VAC | Valderejo | 42.879 | -3.214 | 1102 | 20 | C | 2013-2015 | GV+ v3.2 | ICTA-ICP |
| WAO | Weybourne | 52.950 | 1.122 | 20 | 10 | C | 2007-2015 | GV+ v3.2 | UEA |
| WES | Westerland | 54.930 | 8.320 | 12 | 9 | C | 2006-2015 | WDCGG | UBA |

**Table 1.** Observation sites used in the inversions. Datasets with in-situ continuous (C) as well as flask (F) measurements were taken from GLOBALVIEWplus ObsPack, WDCGG, and the EU-funded projects CarboEurope-IP, GHG-Europe and ICOS preparatory phase (all indicated as preICOS).





derived EVI (enhanced vegetation index) and LSWI (land surface water index). Model parameters were optimised for
Europe using eddy covariance measurements made during 2007 from 47 sites (Kountouris et al., 2015). The VPRM
simulation used here has been produced at a 0.25 degree spatial and hourly temporally resolution.

The mean seasonal cycle and the inter-annual variability of these NEE simulations are shown in Figure 2. Among the notable
features is the annual mean NEE of VPRM, which is much lower ($\approx$-1.1 PgC/year) than that of the three other models (ranging
from -0.1 to -0.4 PgC/year). VPRM is known to produce a too large uptake (Oney et al., 2017), which can be explained by
the optimisation of this diagnostic model against flux measurements from one year. The year to year variations of the annual
budget are significant ($\approx$0.1 PgC/year) but not always in phase between the four models. For the mean seasonal cycle, the peak
to peak amplitude differs significantly between the models with the smallest amplitude obtained with LPJ-GUESS (around 0.4
PgC/months) and the largest with ORCHIDEE (around 0.8 PgC/month). Another visible feature is the phasing of the seasonal
cycle in LPJ-GUESS with an earlier $CO_2$ peak uptake than the other three models (May versus June) and a peak release in
August. This phase difference has already been described by Peng et al. (2015).

### 3.2.2   Anthropogenic emissions

The anthropogenic emissions from combustion of fossil fuels and biofuels, and from cement production are based on a pre-
release of the EDGARv4.3 inventory for the base year 2010 (Janssens-Maenhout et al., 2019). This specific dataset includes
additional information on the fuel mix per emission sector (Janssens-Maenhout, pers. comm.) and thus allows for a temporal
scaling of the gridded annual emissions for individual years (2006–2015) according to year-to-year changes of fuel consump-
tion data at national level (bp2, 2016), following the approach of Steinbach et al. (2011). A further temporal disaggregation
into hourly emissions is based on specific temporal factors (seasonal, weekly, and daily cycles) for different emission sectors
(Denier van der Gon et al., 2011). The seasonality and inter-annual variability of this anthropogenic emissions prior are also
reported in Figure 2 (in black).
Agricultural waste burning is already included in the version of the EDGAR v4.3 anthropogenic emission inventory that we
are using. Also, large scale biomass burning emissions are negligible in Europe (of the order of 0.01 PgC/year), therefore we
decided that no extra biomass burning emission data set should be used in the inversions. Nevertheless, two models (LUMIA
and FLEXINVERT) included a prescribed biomass burning source, based on the Global Fire Emission Database v4 (Giglio
et al., 2013).

### 3.2.3   Ocean fluxes

The role of the ocean flux in causing spatial $CO_2$ gradients between stations at the European scale is very minor in regard to
the magnitude of other fluxes (below -0.1 Pgc/year). Therefore modelling groups were free to choose which ocean fluxes to
use.
   Two groups (LUMIA and FLEXINVERT+) used ocean fluxes from the CarboScope surface-ocean $pCO_2$ interpolation
(oc_v1.6 and oc_v1.4 respectively) (Rödenbeck et al., 2013). The CarboScope interpolation provides temporally and spa-

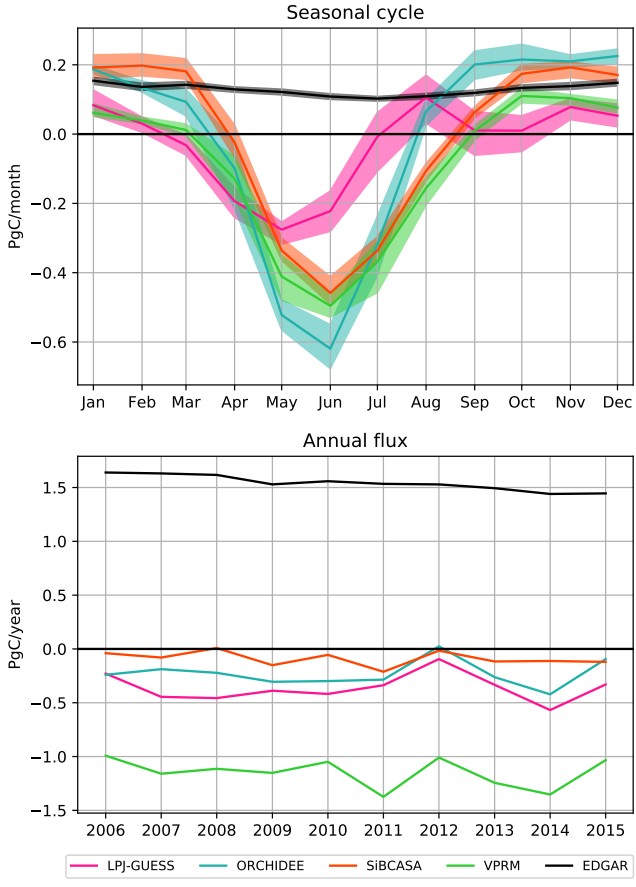

**Figure 2.** Seasonal cycle (top) and inter-annual variability (bottom) of the prior NEE (colored lines/shades) and prescribed anthropogenic flux (black) used in the inversions, for geographical Europe (see definition in Section 3). The solid lines in the upper plot represents the mean seasonal cycle over the 10 years of the study, while the shaded envelopes show the min/max values over the same period.

tially resolved estimates of the global sea-air $CO_2$ flux. Fluxes are estimated by fitting a simple data-driven diagnostic model of ocean mixed-layer biogeochemistry to surface-ocean $CO_2$ partial pressure data from the SOCAT database. NAME-HB used a climatological prior from Takahashi et al. (2009), which is based on a climatology of surface ocean $pCO_2$ constructed using measurements taken between 1970 and 2008. The CarboScope-Regional inversion used an ocean flux estimate taken from the

Mikaloff Fletcher et al. (2007) global oceanic air-sea $CO_2$ inversion and CarbonTracker Europe optimized prior fluxes from the ocean inversion of Jacobson et al. (2007). Finally, PYVAR-CHIMERE used a null ocean prior, but allowed the inversion to adjust it.



### 3.3 Inversion systems

The six inversion systems encompass a wide range of mesoscale regional transport models (with both Lagrangian and Eulerian models) and of approaches for the inversion (variational, ensemble and MCMC methods). The systems also differ by the definition of the boundary conditions, the selection of the observations to be assimilated, the definition of the control vector and the parameterisation of uncertainty covariance matrices. Table 2 presents an overview of the participating systems characteristics.

#### 3.3.1 Transport models

Four out of the six inversions rely on transport from Lagrangian transport models (LUMIA, FLEXINVERT+, Carboscope-Regional and NAME-HB), while the two others (PYVAR-CHIMERE and CTE) rely on Eulerian models. This distinction between Eulerian and Lagrangian models is important as it has practical consequences on how the boundary conditions (initial $CO_2$ concentrations and impact of $CO_2$ fluxes outside the regional domain) can be imposed, but also on how the sensitivity to surface fluxes is defined.

In Eulerian models, the atmosphere is represented by a 3D grid (latitude, longitude and height). The $CO_2$ concentration is defined at each grid point and is altered at each time step by the $CO_2$ sources and sinks (i.e. the inversion control vector) in the surface layer, and by the air mass exchanges between the grid cells (at all layers). Boundary conditions are provided in the form of an initial $CO_2$ field and, when needed (in regional models), as a set of prescribed $CO_2$ concentrations at the edges of the domain. Two inversion systems rely on Eulerian models:

- PYVAR-CHIMERE relies on the CHIMERE model. CHIMERE is a regional Eulerian Chemistry transport model (Menut et al., 2013), forced with ECMWF operational forecasts. The simulations are performed at a horizontal resolution of $0.5°$ and with 29 vertical levels up to 300 hPa, for the exact EUROCOM domain (as described at the beginning of Section 3). Background concentrations are obtained from the transport (by CHIMERE) of a fixed boundary condition interpolated from the CAMS global inversions of Chevallier et al. (2010).

- The CTE inversions rely on the global Eulerian transport model TM5 (Huijnen et al., 2010), driven by air mass transport from the ECMWF ERA-Interim reanalysis. TM5 is here ran at a global resolution of $3° \times 2°$, with a nested $1° \times 1°$ zoom over Europe (21°W-39°E, 12-66°N), and 25 vertical sigma-pressure levels.

In the four other systems, Lagrangian transport models are used to compute, for each observation, a response function (footprint), i.e. a Jacobian matrix containing the sensitivity of the observed concentration to surface fluxes. The change in $CO_2$ concentrations resulting from the surface fluxes are simply the dot product of each footprint by the corresponding (slice of) the flux vector.

Lagrangian models typically simulate the dispersion backwards in time from each observation point of a large number of air trajectories (the approaches to do so differ between the models). The aggregated residence time of the air in individual surface grid boxes is taken as a proxy for the sensitivity of the observation point to surface processes in each of these grid boxes. The footprints are necessarily limited in time (each covers a period of at most a few weeks before each observation), and in



most instances also in space (unless a global Lagrangian model is used). A "background" term representing the contribution of fluxes outside the space/time domain of the footprint needs to be added to represent the total modelled $CO_2$ concentration.

The four inversions relying on such pre-computed footprints differ by the actual Lagrangian models used, but also by the approach used to compute the footprints (the definition of the surface layer) and by the type of background information used:

- The CarboScope-Regional system (Kountouris et al., 2018a) relies on footprints from the STILT model (Lin et al., 2003).
  STILT footprints are computed for the exact EUROCOM domain, at a horizontal resolution of $0.25°$, and at a hourly temporal resolution, and they cover a period of 10 days prior to each observation. STILT is driven by short-term forecasts of the ECMWF-IFS model at $0.25°$ resolution. The surface layer (up to which surface fluxes are mixed instantaneously) is defined as half the height of the planetary boundary layer, at any given time. The background concentrations are computed directly at each observation site by a global, coarse resolution CarboScope $CO_2$ inversion (Rödenbeck et al.,
  2003), following the 2-step approach described in Rödenbeck et al. (2009).

- In LUMIA, footprints covering the EUROCOM domain at a $0.5°$, 3-hourly resolution were generated with the FLEX-PART 10.0 model (Pisso et al., 2019), driven by ECMWF ERA-Interim meteorology. The footprints cover a period of seven days prior to each observation and the surface layer is defined as the atmosphere below 100 m a.g.l.. The background concentrations are also computed following the Rödenbeck et al. (2009) approach, but this time a global
  TM5-4DVAR inversion is used for computing the background concentrations (Monteil and Scholze, 2019).

- The FLEXINVERT+ inversion (Thompson and Stohl, 2014) also relies on footprints from the FLEXPART model, but driven by ECMWF operational forecasts. In contrast to CarboScope-Regional and LUMIA, the footprints are computed globally, on a $0.5°$ hourly grid, and cover a period of five days before each observation. Since the footprints are global, the background (from the perspective of the transport model) results only from the transport to the observation sites of
  the initial $CO_2$ distribution (i.e. the $CO_2$ distribution at the start of the period covered by each footprint). This initial concentration is calculated as a weighted average of a global $CO_2$ distribution sampled where and when the FLEXPART trajectories are terminated, and this global $CO_2$ distribution is based on a bivariate interpolation of observed $CO_2$ mixing ratios from NOAA sites globally, with monthly resolved fields. Note that for this system, the domain of the transport model is larger than that of the inversion itself.

- 315 The NAME-HB system (White et al., 2019) uses footprints from the NAME Lagrangian particle dispersion model. NAME is driven by 3-hourly meteorology from the UK Met Office's Unified Model (Cullen, 1993), at a spatial resolution which changes in time and is $0.233°$ latitude by $0.352°$ longitude before mid 2014. The footprints are defined on a large regional domain, ranging from $97.9°W$; $10.729°N$ to $39.38°E$; $79.057°N$, with a spatial resolution of $0.233° × 0.352°$ (it covers the eastern half of North America, Europe and the Northern half of Africa). The footprints are computed for a
  period of 30 days before each observation, at a 2-hourly temporal resolution in the first 24 hours, and the remaining 29 days are integrated. The surface layer is defined as the layer below a height of 40 m. The background is derived from a global $CO_2$ simulation with the MOZART transport model (Palmer et al., 2018). The MOZART $CO_2$ field is sampled at the time when and location where the NAME trajectories leave the NAME domain.


| Inversion systems (references) | PYVAR-CHIMERE (Broquet et al., 2011; Fortems-Cheiney et al., 2019) | LUMIA (Monteil and Scholze, 2019) | FLEXINVERT+ (Thompson and Stohl, 2014) | CarboScope-Regional (Kountouris et al., 2018a) | CarbonTracker Europe (Peters et al., 2010; van der Laan-Luijkx et al., 2017) | NAME-HB (White et al., 2019) |
|---|---|---|---|---|---|---|
| Institute | LSCE | Lund University | NILU | MPI-BGC-Jena | Wageningen Univ. | Univ. Bristol |
| Method | Variational | Variational | Variational | Variational | EnKF | MCMC |
| Transport model | CHIMERE (Eulerian) | FLEXPART (Lagrangian) | FLEXPART (Lagrangian) | STILT (Lagrangian) | TM5 (Eulerian) | NAME (Lagrangian) |
| Meteorological forcing | ECMWF ERA operational forecasts | ECMWF ERA-Interim reanalysis | ECMWF operational forecasts | Short term forecasts of ECMWF-IFS at 0.25° resolution | ECMWF ERA-Interim reanalysis | UK Met Office's Unified Model (Cullen, 1993) |
| Background | Prescribed At domain edge from a CAMS LMDZ inversion | Prescribed At obs. location from a TM5-4DVAR inversion | interpolation of NOAA data + transport of prescribed fluxes outside the EUROCOM domain | Prescribed At obs. location from a global CarboScope inversion | None (global inversion) | Optimised at domain edge, from a MOZART simulation prior |
| transport and inversion domain | 31.5°N to 74°N; 15.5°W to 35°E | 33°N to 73°N; 15°W to 35°E | Global transport, inversion on a 30°-75°N, -15°-35°E domain | 33°M to 73°N, 15°W to 35°E | Global, zoom over Europe (21°W-39°E, 12-66°N) | 10.729°N to 79.057°N; 97.9°W to 39.38°E |
| Inversion spatial resolution | 0.5° × 0.5° | 0.5° × 0.5° | PFTs × countries | 0.5° × 0.5° | 1° × 1° over Europe, 3° × 2° globally | Large regions × PFTs |
| Inversion temporal resolution | 6 hours | 1 month | 12 hours | 3 hours | weekly | variable (max 1 day) |
| Prior estimate of NEE | ORCHIDEE | LPJ-GUESS | ORCHIDEE | VPRM | SiBCASA | ORCHIDEE |
| Correlation (spatial, temporal) scales of the prior uncertainty | 200 km, 1 month | 200 km, 1 month | No spatial correlation between PFT/country regions, 1 month | 100 km, 1 month | 200 km with no correlation between different PFTs, 5 weeks | No correlations (large regions) |
| Ocean fluxes | part of the control vector (6 hour and 0.5° resolution), null prior | Prescribed (Rödenbeck et al., 2013) | Prescribed (Rödenbeck et al., 2013) | Prescribed (Mikaloff Fletcher et al., 2007) | Optimised (Jacobson et al., 2007) | Prescribed (Takahashi et al., 2009) |

**Table 2.** Overview of the inverse modelling systems and configuration of the inversions





### 3.3.2 Inversion approaches

Four out of the six systems (PYVAR-CHIMERE, LUMIA, CarboScope-Regional and FLEXINVERT+) implement a variational inversion approach, in which the minimum of the cost function $J(\mathbf{x})$ (Eq. 1) is searched for iteratively. The CTE inversion (Peters et al., 2007; van der Laan-Luijkx et al., 2017) employs an ensemble Kalman smoother with 150 members and a 5-week fixed-lag assimilation window. The NAME-HB inversion uses the MCMC method (Rigby et al., 2011; Ganesan et al., 2014; Lunt et al., 2016; White et al., 2019). In short, this method samples the parameter space and proposals for parameter

values are accepted or rejected according to some rules based on the likelihood of the proposal.

Regardless of the inversion technique used, all the groups were asked to provide optimised NEE fluxes at a monthly, $0.5°$ resolution on the EUROCOM domain. However, the precise control vector optimised in some of the inversions differ from this requested product:

- In PYVAR-CHIMERE, the NEE is optimised at a 6-hourly resolution on each grid cell (on the standard EUROCOM

grid), starting from a prior NEE estimate from the ORCHIDEE model (See Section 3.2.1). In addition, the inversion also adjusts the ocean flux estimate, starting from a null prior. The prior uncertainty for each control vector element is proportional to the respiration in the corresponding grid cell (according to the same ORCHIDEE simulation) and further scale to obtain an average uncertainty at the $0.5°$ and 1 day scale of 2.27 $\mu$mol.CO$_2$/m$^2$/s (after Kountouris et al. (2018a)).

- The LUMIA inversion controls the NEE fluxes monthly, on the standard EUROCOM grid, starting from prior NEE from the LPJ-GUESS model. The prior uncertainty is set to 50% of the prior control vector (i.e. the prior NEE), with a minimum uncertainty of set to 1% of the grid point with the largest uncertainty, to avoid zero-uncertainty when NEE is close to zero. The decadal inversion was decomposed in ten 14-month inversions, from which the first and last month were not used.

- In the CarboScope-Regional system, the NEE fluxes are optimised 3-hourly at a $0.5°$ resolution in the EUROCOM domain, based on a prior NEE estimate from the VPRM model. In addition, the control vector contains a bias term, which scales uniformly the map of annual total respiration. The uncertainty on the prior NEE is set to a uniform value of 2.27 $\mu$mol.CO$_2$/m$^2$/s and the uncertainty on the bias term is adjusted so that the total uncertainty integrated over the domain is 0.3 PgC/year. The setup is identical to the "BVR" case in Kountouris et al. (2018a). The decadal inversion

period was divided in three periods (2006-2007, 2008-2011, and 2012-2015).

- FLEXINVERT+ controls the NEE per country $\times$ Plant Functional Type (116 control variables per time step across Europe, with PFTs based on those in the CLM model). The fluxes are optimised for 6-hourly periods (0-6,6-12,12-18,18-0 local time), averaged over five days. The prior NEE flux is based on the ORCHIDEE simulation described in Section 3.2.1, and the uncertainties are set proportional to this prior NEE. The transport model in FLEXINVERT+ is

global, therefore the flux estimates used in the inversions are defined over the entire globe. However, the inversion only adjusts NEE within the EUROCOM domain.





– In NAME-HB, the domain of NAME has been split into eight boxes: four "background" boxes outside the EUROCOM domain, and four "foreground" boxes within the EUROCOM domain. The latter were further divided based on a PFT map used in the JULES vegetation model (Still et al., 2009), which includes six PFTs. The inversion optimizes separately the gross primary production (GPP, i.e. the uptake of carbon by plants) and the heterotrophic respiration (TER, with NEE=GPP+TER). The flux components are optimized at a variable temporal resolution, with a maximum resolution of one day (see White et al. (2019) for further details). The oceanic flux is prescribed (based on the Takahashi et al. (2009) climatological pCO$_2$ estimate), but the background concentrations are part of the control vector and are therefore adjusted during the inversion. Therefore, there are 56 elements in the control vector, 4 elements to optimise the background concentrations, 4×2 elements to optimise the "background" regions for each of GPP and TER, 4×5 elements for the PFT-regions for GPP (as one of the 6 PFTs is not applicable to GPP) and finally 4×6 elements for the PFT-regions for TER. The uncertainties are set to 100% of the prior for GPP and TER, and to 3% of the initial value for the background terms.

– in CTE, the NEE and ocean fluxes are optimised globally on a weekly time resolution in a 5 week lagged window. The global domain is split in 11 TRANSCOM regions, which are further decomposed in ecoregions corresponding to 19 ecosystem types. The fluxes are optimised on 1x1 degrees resolution for the Northern Hemisphere land regions, and by ecoregion and ocean region for the rest of the world. The prior NEE is taken from the SiBCASA simulation described in Section 3.2.1 and the prior oceanic flux is based on Jacobson et al. (2007).

In the three systems that optimise NEE at the pixel scale (LUMIA, PYVAR-CHIMERE and CarboScope-Regional), the spatial resolution of the control vector is in practice further limited by the use of distance based spatial and temporal covariances in the flux covariance matrices (**B** in Equation 1), which in effect smoothes the results by preventing the inversion from adjusting neighbouring pixels totally independently. The values of 100 km (CarboScope-Regional) and 200 km (PYVAR-CHIMERE and LUMIA) used for the spatial covariance lengths correspond well to the diagnostics of comparisons between the ecosystem simulations and flux eddy covariance measurements (Kountouris et al., 2018a). These systems and FLEXINVERT+ also assume temporal error covariances of one month at each grid cell.

The NAME-HB and FLEXINVERT+ inversions only control a limited number of PFTs in each region, which means that pixels in the same region and corresponding to the same PFT have a correlation coefficient of 1. Finally, CTE follows an intermediate approach. The flux uncertainties of Northern Hemisphere land pixels within a same ecoregion are correlated following with a variable spatial covariance length to reflect the observation network density (200 km in Europe), and the uncertainties of grid boxes corresponding to different ecoregions are assumed uncorrelated. For the rest of the world, the uncertainties are coupled within each TRANSCOM region decreasing exponentially with distance. The chosen prior standard deviation is 80% on land parameters, and 40% on ocean parameters (van der Laan-Luijkx et al., 2017).





### 3.3.3 Observation vectors and errors

All the inversions use observations from the stations listed in Table 1. Each participant was, however, free to refine their
selection of observations (both in terms of number of sites assimilated and of data selection at each site) to adapt it to the skills
of their own inversion system. In practice, five of the six inversions used data from nearly all the observation sites. NAME-HB
used only a restricted list of 15 sites (see Figure SI1)

Most of the systems assimilate instantaneous or 1-hour averages of the measurements, taken, when there are several vertical
levels of measurements, at the top level of the stations, as it is the least sensitive to very local surface fluxes. NAME-HB
assimilates 2-hourly observations (average of the observed concentrations in each 2-hourly interval). Due to the traditional
limitations of transport models in terms of representation of the orography and simulation of the vertical mixing (Broquet et al.,
2011), most of the systems use observations at low altitude sites during the afternoon only, and observations at high altitude
sites during night time only (vertical gradients of $CO_2$ near to the surface are notoriously difficult to simulate accurately, so
observations when the vertical gradients are expected to be the lower are preferred)

– PYVAR-CHIMERE assimilates 1-hour averages of the continuous or flask measurements over specific time windows
that depend on the altitude of the stations above the sea level. The selection window is 12:00-18:00 UTC time for
stations below 1000 m a.s.l. and 0:00-6:00 local time for stations above 1000 m a.s.l. (following the analysis and choices
by Broquet et al. (2011)). The observation errors are set-up as a function of stations, of the height of the station level
above the ground and of and season, following the estimates by Broquet et al. (2011, 2013), based on comparison of
simulations and measurements of Radon). Their standard deviation for the 1-hour averages ranges from 3 to 17 ppm.

– In LUMIA, observations from sites with continuous observations are selected based on the
"`dataset_time_window_utc`" flag in the metadata of the observation files. That corresponds, for most sites,
to a 11:00 to 15:00 UTC time range, and to a 23:00 to 03:00 UTC time range for mountain sites. At sites with only
flask observations, all samples were used. The observation uncertainties are defined as the quadratic sum of the measure-
ment uncertainties, of the uncertainty of the foreground transport model (i.e. FLEXPART) and of the uncertainty on the
background concentrations. The measurement uncertainties are taken from the data files when available, and a minimum
uncertainty of 0.3 ppm is enforced. Foreground transport model uncertainties are computed by performing two similar
forward model runs, with TM5 and LUMIA (i.e. FLEXPART + background concentrations from TM5), configured such
that the only difference is the model used to compute the transport within the EUROCOM domain. The uncertainties on
the background concentrations are set as the standard deviation of the vertical profile of background $CO_2$ concentrations
around each observation (see Monteil and Scholze (2019) for details about the approach). The combined uncertainty is
on the order of 4 ppm, on average.

– CarboScope-Regional assimilates observations, between 11:00 and 16:00 UTC for tall-towers, ground-based or coastal
stations, and from 23:00 to 04:00 UTC for mountain stations (the time intervals refer to the beginning of the observation
hour). A base representation error of 1.5 ppm was assumed for tall towers, coastal and mountain. For ground based





continental sites it was raised to 2.0 ppm, and to 4 ppm for Heidelberg, which is in a urban environment. For sites that provide hourly observations, an error inflation was applied (e.g. for tall towers: 1.5 ppm $\times \sqrt{6 \text{ obs/day} \times 7 \text{ day/week}} =$ 9.7 ppm).

– In FLEXINVERT+, observations were assimilated hourly between 12:00 and 16:00 local time for sites below 1000 m.a.s.l. and between 00:00 and 04:00 for sites higher than 1000 m.a.s.l. The observation uncertainties are calculated as the quadratic sum of the measurement errors (with a minimum of 0.5 ppm), the uncertainty on the initial mixing ratio, assumed to be 1 ppm and the contribution of uncertainties in the fossil fuel emission estimates and in the NEE fluxes from outside the domain, both transported by FLEXPART to the observation sites. The total observation-space uncertainties typically range between 1 and 3 ppm.

– In NAME-HB, observations are filtered based on a combination of two metrics. One is the ratio of the NAME footprint magnitude in the 25 grid boxes closest to the measurement site. If this ratio is high it indicates that a large proportion of the air arriving at a measurement site is from very local sources and may not be resolved by the model. The second metric is the lapse rate modelled by NAME, which is the change of temperature with height and is a measure of atmospheric stability. A high lapse rate suggests very stable atmospheric conditions and may also indicate that there is a lot of local
influence on the measurement. With these criteria, some data outside the usual daytime time constraints can be included and daytime data that is not collected during favourable conditions can be removed. In practice however, most of the data included is during the daytime. The measurement uncertainties are taken from the data providers and averaged over the month for each measurement site to give a fixed monthly value. The observation uncertainty is adjusted during the inversion but initially it is the sum of the average measurement uncertainty and a model uncertainty of 3 ppm.

– In CarbonTracker-Europe, flags from data providers are used to screen for representative observations (usually equivalent to the afternoon hours for typical sites and night time hours for mountain sites). A model-data mismatch based on the station category (tower, flask, etc.) is assigned to each site, accounting for both measurement errors and modelling errors at that site. If the difference between the forecast and observation is greater than three times that assigned model-data mismatch, the observations is not used in the inversion.

The range of uncertainties varies a lot across the systems, and can range from one up to tens of ppm. It reflects the different types of coupling between global and regional transport models, and the different range of diagnostics available for each group to quantify their uncertainties. The precise impact of these differences in prescribed observation uncertainties will be analysed in a follow up study.





## 4 Results

### 4.1 Fit to the observations

Before presenting the posterior NEE from the six inversion systems we first briefly analyse the reduction of the misfits to the $CO_2$ observation assimilated by each inversion. The aims are to first check that all inversions actually reduce the observation misfits (which is a basic diagnostics of atmospheric inversions) but also to determine whether some sites are particularly problematic for some or all of the inversions.

For each inversion, a comparison of the prior and posterior bias and root mean square (RMS) differences (denoted RMS errors i.e. RMSE) between the time series of measured and simulated data computed at each assimilated site, and between the full sets of measured and simulated data, is shown in Figure 6. For each inversion, the observation sites are sorted according to the corresponding prior bias. The expectation in this analysis is that all the systems should show a reduction of the misfits (both in terms of mean bias and RMSE), which indeed is what happens. Ideally the posterior misfits should also be close to unbiased.

All inversion satisfy that expectation, and lead to a posterior bias lower than 0.5 ppm (the prior biases are also close to zero for LUMIA, CTE and FLEXINVERT+), and all lead to a net reduction of the spread of the residuals, with RMSE ranging from 2.9 ppm (CTE) to 5.4 ppm (CarboScope-Regional). NAME-HB and CarboScope-Regional both start from a relatively larger average negative prior bias (respectively -0.94 and -0.71 ppm), whereas the other systems all start from prior biases ranging between -0.26 ppm (PYVAR-CHIMERE) and 0.27 ppm (CTE). In the case of CarboScope-Regional, this is easily explained by the substantially larger prior $CO_2$ sink in the VPRM prior (Section 3.2.1), while since NAME-HB uses the same ORCHIDEE prior as other inversions (PYVAR-CHIMERE, FLEXINVERT+), its prior bias must have a different origin (background, transport or oceanic flux). Note that the NAME-HB inversion only covers a reduced 5-years period (2011-2015), which limits its comparability with the other inversions.

At the site scale, the decrease of the misfits is rather moderate, up to 30% but, mostly below 20%, without a clear distinction between low altitude and high-altitude sites or between the models. Each inversion occasionally leads to local degradation of the fit (increase of the bias or RMSE). Such degradation can occur when the inversions do not have enough independent degrees of freedom to reconcile contradictory constraints from several sites. This can be because the spatial resolution of the control vector is too low compared to the density of sites, in which case it does not necessarily impact negatively the accuracy of the solution. But it can also be an indication that a site is not well represented by the transport model and could as a consequence introduce a local bias in the posterior flux. Some sites tend to be systematically misrepresented by the inversions (including in the posterior step), in particular those in the vicinity of large urban areas (with large anthropogenic emissions), such as HEI and GIF. Note that this is accounted for in several of the inversions, but not all, by inflating the model representation errors (which allows the model to degrade the fit to the observations, at a low "cost"). Besides these two sites, if does not appear the distribution of the fits is systematic. Especially, there is no major difference between the representation of mountain-top (with night-time observations assimilated) and plain sites.





Error statistics computed on 1-month and 1-year averages of the observations (Table SI1) show larger RMS error reductions in most models (up to 42%, and 33% respectively on the whole observation ensemble, in FLEXINVERT+). However, the RMS error reduction for annual averages are generally smaller than that for monthly averages, which suggests that results at the monthly scale are likely more robust than at annual to multi-annual scales.

This comparison of the residuals is an important technical diagnostics, but does not indicate how realistic the posterior fluxes are, and should not be interpreted as a ranking of the inversions. Especially, a good posterior representation of the observations is only a sign that the inversion had enough independent degrees of freedom to match the observed concentrations, but does not mean that the observations are sufficient to robustly constrain the control vector, or that the underlying transport model is accurate.

## 4.2 Posterior European-scale NEE

The monthly prior and posterior NEE from the six inversions, integrated over the whole European domain (as defined in Section 3) and over the 10-year period of the intercomparison are displayed in Figure 3. The dominant feature in Figure 3 is the systematic differences between the seasonal cycles, i.e. each inversion shows a similar pattern in the seasonal cycle (large/small amplitude or timing of peak values) for each year of the simulation period.

Overall, the posterior fluxes remain within or close to the range of values defined by the different priors. In the Sections 4.2.1 and 4.2.2, we compare the prior and posterior mean fluxes and their variability, at the annual and monthly time scales. In Section 4.2.3 we have a first look at the sub-continental scale. Results at the grid scale are provided for completeness in SI, but are not further discussed in this paper.

### 4.2.1 Long term mean and variability of the annual NEE budget

Prior and posterior estimates of the annual budgets of the NEE over the European domain, as well as their mean and standard deviation over the inversion period are reported in Figure 4.

We find an ensemble mean posterior estimate of the 10-year average NEE of -0.19 PgC/year, with values ranging from a net source of 0.28 PgC/year (PYVAR-CHIMERE) to a net sink of -0.61 PgC/year (CarboScope-Regional). Besides PYVAR-CHIMERE, only the NAME-HB system finds the European ecosystems on average to be a net source of $CO_2$ to the atmosphere over our simulation period. The CTE inversion yields an almost neutral NEE budget (+0.02 PgC/year), and LUMIA and FLEXINVERT+ both derive a sink of ≈-0.4 PgC/year. Overall, the range of estimates from our inversions (0.9 PgC/year) is slightly narrower than that of the priors (1.06 PgC/year between VPRM and SiBCASA).

The last column of the Table in Figure 4 shows the standard deviation of each annual NEE estimate, which we use as metrics for their inter-annual variability (IAV). It is lower than 0.17 PgC/year for almost all the estimates; FLEXINVERT+ is an exception with a annual NEE standard deviation of 0.31 PgC/year. The differences in IAV are therefore, with the possible exception of FLEXINVERT+, only a small contributor to the posterior range of annual budgets (the standard deviation of the ensemble is, on average, 0.34 PgC each year for the posterior NEE). This further highlights that differences between annual budgets of the inversions are primarily driven by the differences in long-term average, and that the latter is not robustly constrained in our


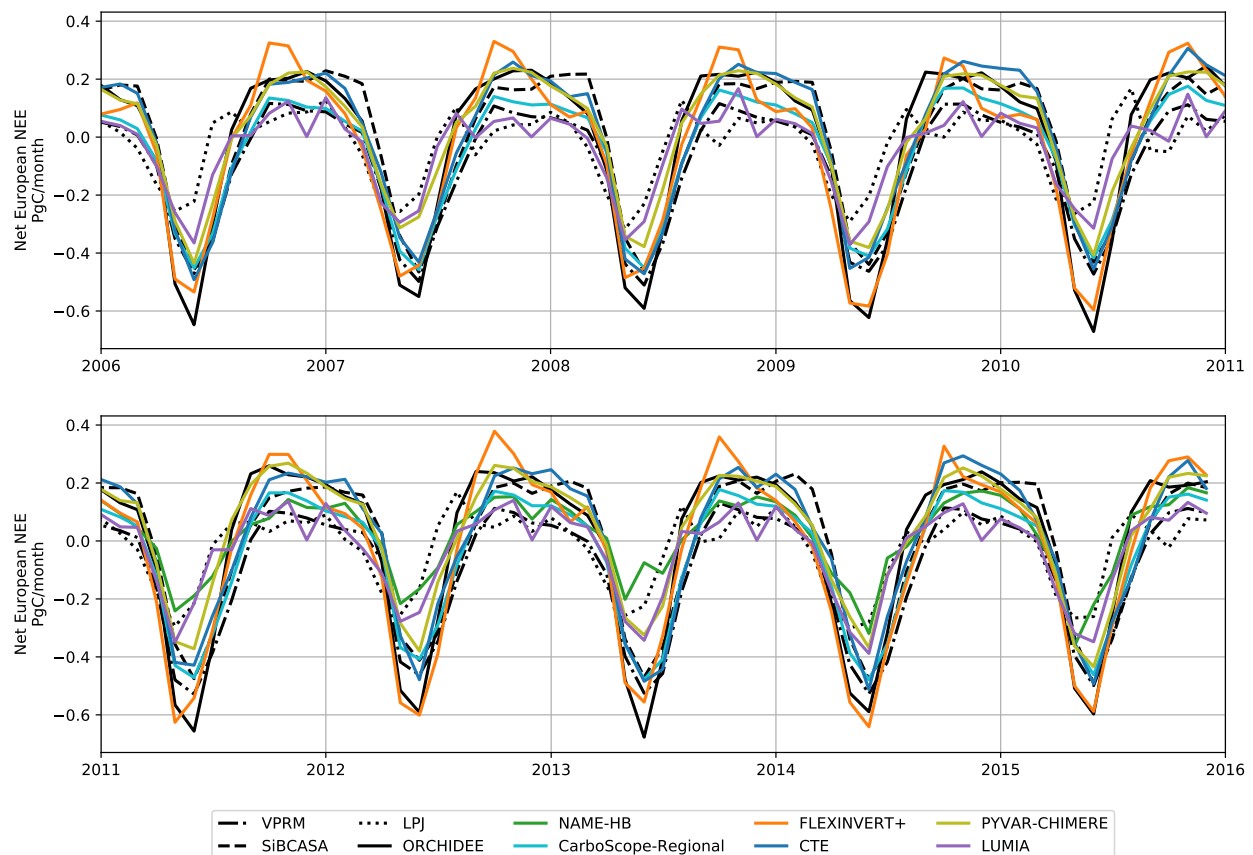

**Figure 3.** Monthly posterior fluxes, aggregated on the entire domain.

set of regional inversions. In the case of the CarboScope-Regional inversion, an obvious source for an offset from the other inversions is the prior flux from VPRM, which is much more negative than the other three priors. However, the differences between the three inversions using the NEE field from ORCHIDEE as a prior flux (PYVAR-CHIMERE, FLEXINVERT+ and NAME-HB) show that the biases between prior estimates can, at best, only partially explain the offsets in posterior estimates.

The annual anomalies of NEE are compared in Figure 5, and the colors of the cells in Figure 4 also scaled to these anomalies
(with the long term mean of each estimate taken as a reference). The ensemble spread of the posterior anomalies is generally much larger than that of the prior, although one system (FLEXINVERT+) is contributing the most to such spread (i.e. in 2009 and 2014 for instance). The spread also strongly varies from year to year, from a minimum spread of 0.16 PgC/year in 2012 to a maximum of 0.67 PgC/year in 2009. In order to provide a metrics less sensitive to potential model outliers, the medians of the prior (blue) and posterior (red) anomalies are shown in the Figure 5.
The difference between the median prior and posterior anomalies is large at the start of the inversion period, with a median prior flux anomaly of +0.09 PgC/year in 2006 corrected to a median negative anomaly of -0.14 PgC/year by the inversions.





Posterior anomalies in 2006 range between -0.2 and -0.1 PgC/year, except for one inversion which reaches 0.3 PgC/year. Therefore, this correction of the NEE by (most of) the inversions in 2006 seems relatively robust. On the contrary, one clear positive anomaly (≈+0.2 PgC/year) is present in 2012 in the prior fluxes and is further confirmed by all the inversions. While

it is limited to 2012 in the priors, it already starts in 2011 in some of the inversions (PYVAR-CHIMERE and LUMIA) and extends to 2013 in most of the inversions (particularly the three inversions using ORCHIDEE fluxes as prior: NAME-HB, PYVAR-CHIMERE and FLEXINVERT+). The positive 2012 anomaly is also followed by an almost equivalent negative anomaly in 2014 (-0.2 PgC/year in the priors, -0.13 PgC/year in the posteriors) with this time however, a large spread between the posteriors (from -0.55 PgC/year for FLEXINVERT to +0.11 PgC/year for CTE). Another relatively large (0.14 PgC/year)

divergence between the prior and posterior anomalies is found in 2009, but because of the very large spread in the inversion ensemble that year, we do not consider it very robust. For most of the other years during this 10-year period, the median for both the prior and posterior estimates of the NEE generally indicate small anomalies, but individual inversions can diverge a lot from the rest of the ensemble of posterior estimates, like FLEXINVERT+ before 2010 and in 2014.

In summary, a few robust features in terms of flux anomalies seem to be captured by the inversion ensemble (the 2006 and

2012-2014 anomalies), but the analysis of their drivers is complicated by the aggregation over the large spatial and temporal scales: the observation network is not homogeneous, and the inversions may constrain some regions of the domain or times of the year better than others. In the following sections (4.2.2 to 4.2.3) we analyse the inversion results at finer temporal and spatial scales.

### 4.2.2  Seasonal variability of NEE

The mean monthly posterior NEE estimates for the six inversions together with the prior fluxes are shown in Figure 7. At first glance, the spread of the posterior fluxes matches approximately that of the prior estimations, with very similar mean spring (May-June) uptakes, ranging from -0.24 (NAME-HB) to -0.55 PgC/month (FLEXINVERT+) in the posteriors and from -0.28 to -0.62 PgC/month in the priors. Winter posterior emissions are slightly higher (from +0.13 (LUMIA) to +0.32 PgC/month in FLEXINVERT+) than the priors (+0.11 to +0.23 PgC/month). As a result, the median seasonal cycles are also very similar,

with a similar phasing and a seasonal cycle amplitude of ≈ 0.55 PgC.

This similarity between the prior and posterior ensembles hides more significant differences at the level of individual ensemble members. The phasing of the seasonal cycle is very consistent among the inversions, with terrestrial ecosystem becoming a $CO_2$ sink (flux sign switch around April and August and with a peak uptake in June). On the contrary, the bottom-up simulations used as priors have four relatively distinct seasonal patterns (see also Figure 2).

For instance, LPJ-GUESS simulates an early peak $CO_2$ uptake in May, which is not confirmed by the inversions (only NAME-HB yields to a similar peak). LPJ-GUESS simulates a NEE alternating between being a neutral flux and a positive but small (≈0.1 PgC/month) net $CO_2$ source between July and March. This is most of the time outside or at the edge of the range of flux estimates derived from the inversions. The strong peak carbon uptake in June in ORCHIDEE (-0.62 PgC/month in June) clearly exceeds the lower boundary of the posterior ensemble (-0.55 PgC/month, in the FLEXINVERT+ inversion, which is

itself an outlier among the ensemble). The positive NEE found by ORCHIDEE at the end of the summer (0.13 PgC/month



| | 2006 | 2007 | 2008 | 2009 | 2010 | 2011 | 2012 | 2013 | 2014 | 2015 | Mean | Std |
|---|---|---|---|---|---|---|---|---|---|---|---|---|
| PYVAR-CHIMERE | 0.12 | 0.29 | 0.34 | 0.02 | 0.24 | 0.53 | 0.51 | 0.47 | 0.27 | 0.05 | 0.28 | 0.17 |
| CTE | -0.16 | -0.01 | -0.12 | 0.09 | 0.28 | -0.01 | 0.25 | -0.12 | 0.14 | -0.10 | 0.02 | 0.15 |
| FLEXINVERT+ | -0.13 | -0.23 | -0.19 | -1.01 | -0.47 | -0.45 | -0.34 | -0.08 | -0.96 | -0.23 | -0.41 | 0.31 |
| LUMIA | -0.53 | -0.46 | -0.41 | -0.62 | -0.44 | -0.24 | -0.25 | -0.39 | -0.55 | -0.43 | -0.43 | 0.12 |
| NAME-HB | | | | | | 0.09 | 0.35 | 0.40 | 0.07 | 0.17 | 0.22 | 0.13 |
| CarboScope-Regional | -0.75 | -0.82 | -0.52 | -0.54 | -0.39 | -0.71 | -0.40 | -0.66 | -0.76 | -0.53 | -0.61 | 0.15 |
| LPJ-GUESS | -0.23 | -0.45 | -0.46 | -0.39 | -0.42 | -0.34 | -0.09 | -0.33 | -0.57 | -0.33 | -0.36 | 0.12 |
| ORCHIDEE | -0.24 | -0.19 | -0.22 | -0.31 | -0.30 | -0.29 | 0.02 | -0.26 | -0.42 | -0.09 | -0.23 | 0.12 |
| SiBCASA | -0.04 | -0.08 | 0.01 | -0.15 | -0.06 | -0.21 | -0.02 | -0.12 | -0.11 | -0.12 | -0.09 | 0.06 |
| VPRM | -0.99 | -1.16 | -1.11 | -1.15 | -1.05 | -1.37 | -1.01 | -1.24 | -1.35 | -1.03 | -1.15 | 0.13 |
| Median Posteriors | -0.16 | -0.23 | -0.19 | -0.54 | -0.39 | -0.12 | -0.00 | -0.10 | -0.24 | -0.16 | -0.21 | 0.14 |
| Median Priors | -0.23 | -0.32 | -0.34 | -0.35 | -0.36 | -0.31 | -0.06 | -0.30 | -0.49 | -0.23 | -0.30 | 0.11 |

**Figure 4.** Annual NEE budget (positive for a source to the atmosphere and negative for a sink) for the six inversions (upper section of the array), for the four priors (middle section) and median prior and posterior fluxes (lower section). The two last columns show respectively the mean NEE and the standard deviation of the NEE estimate, for each simulation. The colors of the cells indicate the strength and direction of the annual NEE anomaly (respective to the mean of each row, see Figure 5 for the actual values).

in August and September) is also contradicted by the inversions (≈0.04 PgC/month ensemble median in these two months). The phasing of the seasonal cycles in VPRM and SiBCASA are in good agreement with that of the inversions. The winter NEE estimate in VPRM (≈0.07 PgC/month between October and March) is lower than suggested by the inversions (ensemble median of 0.16 PgC/month), and on the contrary, the inversions point to a lower NEE than found by SiBCASA in the first

three months of the year (0.19 PgC/month between January and March, compared to a corresponding ensemble median of 0.09 PgC/month).

The variability of the seasonal cycle for each inversion is illustrated in the right hand plot of Figure 7. The solid lines show the posterior NEE for each year and inversion, and the shaded areas represent the variability of the seasonal cycle inferred by each inversion system. The systematic differences between the inversion systems dominate the picture, and far exceed the

monthly IAV within each inversion during the peak growth period (May-June) and during the fall (October-November). The peak to peak amplitude of the mean seasonal cycle inferred by the different inversions varies between 0.4 PgC/month for NAME-HB and 0.9 PgC/month for FLEXINVERT+

Figure 8 focuses on the monthly anomalies relative to the average seasonal cycle that drive these main differences. The figure provides more insights to explain the IAV of the annual budgets, discussed in Section 4.2.1. For instance, the negative annual





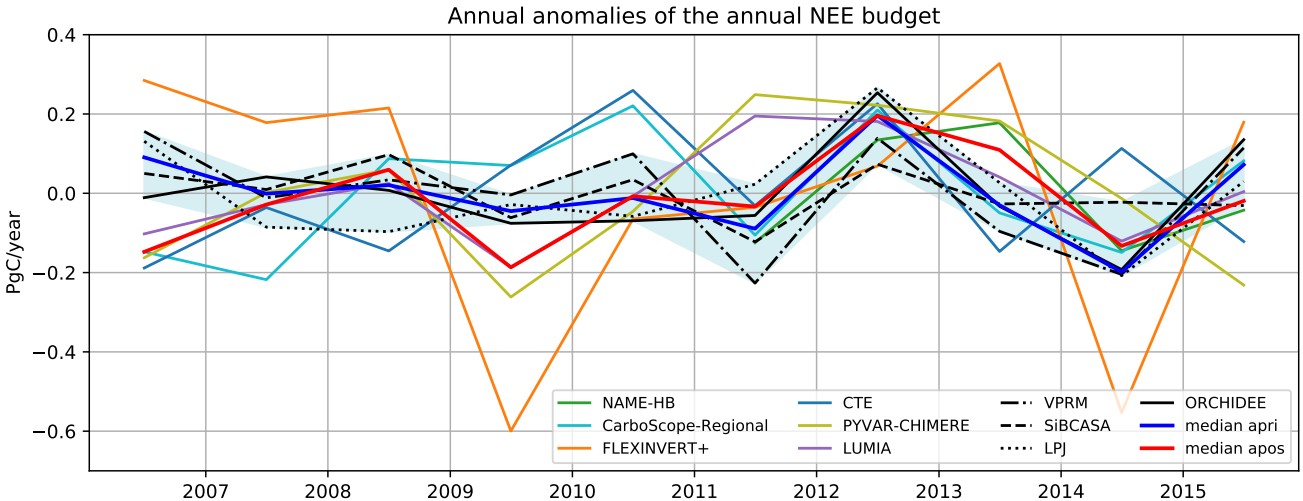

**Figure 5.** NEE anomalies of the six inversion posteriors and of the four priors. The median of the prior anomalies is shown as a thick blue solid line, that of the posteriors is shown as a thick red solid line. The blue shaded area shows the envelope of prior anomalies.

flux anomaly in four inversions in 2006 (i.e. enhanced sink) is explained by a stronger than usual carbon uptake in the summer of 2006. The negative NEE anomaly remains throughout most of the fall and winter of 2006-2007 (up to -0.025 PgC/month during the period May to December 2006) and becomes even more pronounced in March 2007, after which it switches sign. The 2012 anomaly is on the contrary spread over the entire year in almost all the inversions. It is however already well described by the priors, the inversions here provide a confirmation.

In some instances it may be possible to relate these NEE monthly anomalies to climate anomalies. For example, the summer 2006 in Europe was marked by a heat wave lasting for most of the month of July, and was followed by a particularly mild winter, which could explain the relatively stronger carbon sink from June 2006 to May 2007 (Rebetez et al., 2009). However, the size of our domain is assumedly much larger than the spatial extent of most potential climate anomalies, which complicates this type of analysis. We therefore briefly delve in the spatial distribution of the flux adjustments in the following section.

### 4.2.3 Spatial variability

Analysing the spatial variations of the fluxes may reveal robust local signals in areas where the transport models are more reliable and where the observation network is denser. It can also help to better interpret the results in terms of underlying processes in a large region such as Europe where the ecosystems and climate are highly heterogeneous. However, getting robust signals at regional scales is challenging due to the limited spatial resolution of the transport models and to the relative 590 simplicity and large scales of the error correlations used for characterising the prior flux uncertainties. A detailed analysis of the regional signals will be published in a follow-up article. Here, we only provide a brief overview of the spatial distribution of



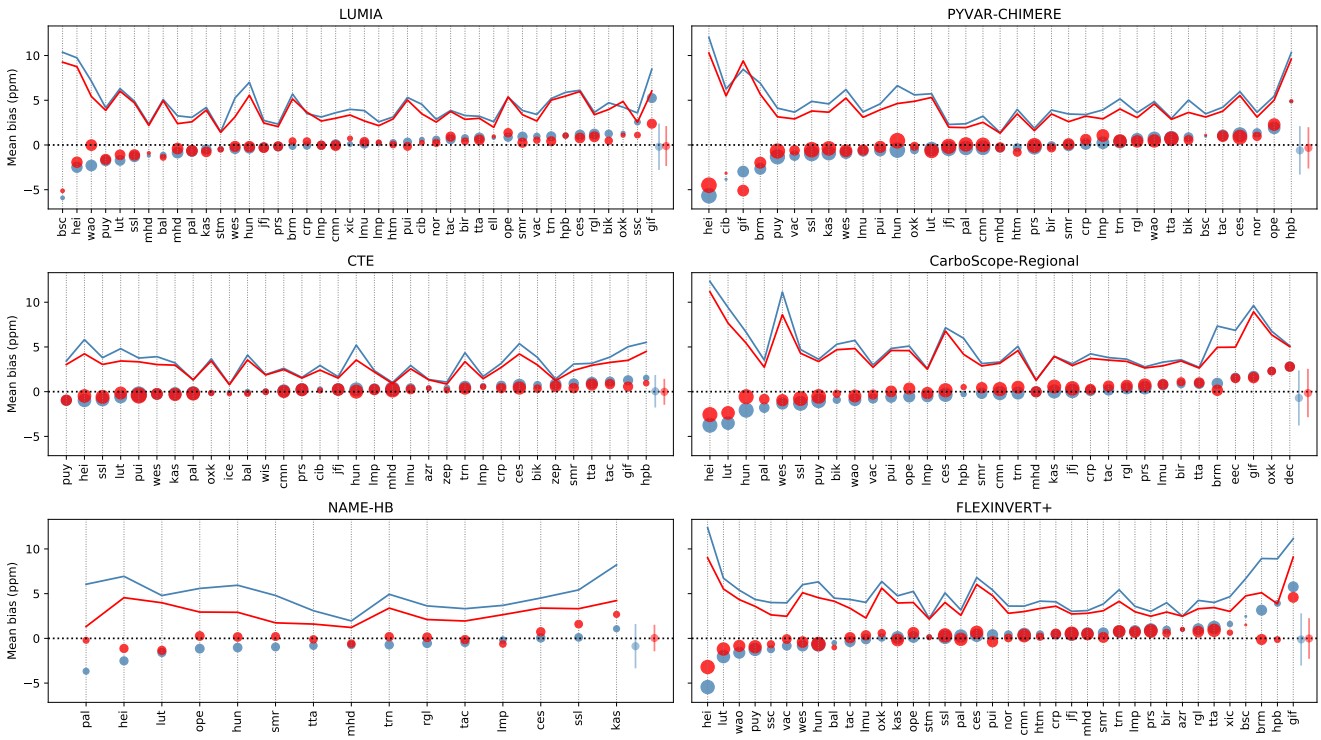

**Figure 6.** Prior (blue) and posterior (red) mean bias (dots) and RMSE (solid lines) at each observation site, for each inversion. The size of the dots is proportional to the number of assimilated observations. The last two points on the right, and their associated error bars represent respectively the mean bias and RMSD taken over the ensemble of assimilated observations.

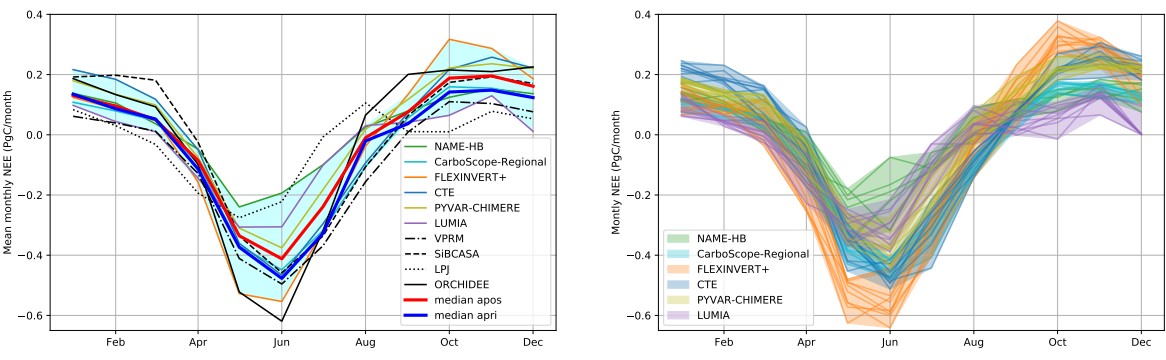

**Figure 7.** left: average seasonal cycles of the prior and posterior estimates. The prior and posterior ensemble median are represented as thick solid lines, and the spread of the posterior ensemble is shown as a shaded area. Right: variability of the seasonal cycle during the ten years of the inversion (the shaded areas represent the range of monthly NEE and the solid lines correspond to the individual years)



**Figure 8.** Monthly prior and posterior anomalies of the seasonal cycle (each simulation compared to its own average seasonal cycle (left hand plot of Figure 7)

the NEE adjustments to provide a first assessment of the potential of regional inversions to analyse subcontinental scale NEE variations and to support the previous analysis of the anomalies at the European scale.

We aggregate the fluxes in four large regions: Northern Europe (Scandinavia, Finland and the Baltic states), Southern Europe
(the Iberian Peninsula, Italy, Greece, Romania and the Balkan states), Western Europe (France, Benelux, UK and Ireland) and Central Europe (the remaining countries, up to the Eastern border of Poland). The regions are pictured in Figure 1. These four regions correspond roughly to four climate zones (Nordic, Mediterranean, oceanic and continental) and exclude parts at the edge of our domain which are not sampled by the observation network (North Africa, Turkey, far east of Europe).





Average regional monthly budgets for both prior and posterior estimates are shown for each region in the upper row of
Figure 9. The figure also shows the median of the prior and posterior ensembles (respectively as thick blue and red lines).
Finally, the spread of the posterior ensemble is highlighted (blue shaded area). The second row of plots show average prior
and posterior regional annual budgets. Some of the systematic differences between the posterior seasonal cycles already noted
at the European domain scale are present in all or most of the regions. This is in particular the case for the lower amplitude
of the NAME-HB seasonal cycle and the autumn positive NEE peak in the FLEXINVERT+ inversion. But others, such as
the positive bias of the PYVAR-CHIMERE posterior (i.e. 0.28 PgC/year, see Figure 4), can be more clearly attributed to one
specific region, like Southern Europe.

### Central Europe

NEE is most robustly estimated in the Central Europe region, which is not surprising because it is the region most densely
sampled by the observation network. The median prior and posterior fluxes are nearly identical, but the spread of the posterior
ensemble is narrower than that of the prior fluxes. In particular, the LPJ-GUESS NEE estimate is clearly outside the range
of posteriors in the summer (it points to a peak uptake of -0.04 PgC/month in June, half of the -0.08 PgC/month ensemble
median).

In terms of net annual budget, the inversions fall in two categories: CarboScope-Regional and LUMIA point to a sink of
-0.12 PgC/year, all the other inversion systems yield a close to zero annual budget. The similarity in the annual budget from
these four inversions is, however, most likely by coincidence because the seasonal distribution of the fluxes is rather different
(FLEXINVERT+ points to a summer uptake 30% larger than that found in the NAME-HB inversion, compensated by larger
winter emissions).

### Western Europe

Western Europe is also well sampled by the observation network, but because of the dominating westerly winds in our domain it
is more sensitive to boundary conditions than the Central Europe region. The spread of the prior fluxes (0.02 to 0.04 PgC/month)
is narrower than in Central Europe and is not further reduced by the inversions. In summer, the NAME-HB inversion suggests a
reduced carbon uptake (-0.02 PgC/month in June, compared to a prior ensemble mean of -0.06 PgC/month), but as mentioned
earlier, this is a systematic feature of that inversion, not specific to Western Europe. In Fall (October to December), two
inversions point to a much stronger positive flux than the priors and the other inversion systems (up to +0.75 PgC/month in
November, double the value of the posterior ensemble mean of +0.35 PgC/month). As a result, there is little convergence
between the annual budgets, which range between a net sink of -0.12 PgC/year (CarboScope-Regional) to a source of 0.06
PgC/year (CTE).

### Southern Europe

The strongest correction to the prior fluxes are obtained in Southern Europe. The median value of the posterior estimates points
to a ≈30% reduction of the summer $CO_2$ uptake compared to the median of the prior fluxes. The spread of the posterior
ensemble is larger than in the other regions (0.03 to 0.1 PgC/month) but the region is also where the spread of the prior interval
is the largest (up to 0.13 PgC/month in July).





The shape of the LPJ-GUESS seasonal cycle is different from that of the other models, with two periods of negative NEE (February-June and October), and a peak carbon flux to the atmosphere in August. For most of the year, it remains outside the range of posterior scenarios, and is therefore not compatible with the atmospheric observations.

The seasonal cycles of the three other prior fluxes are in phase with that of the inversion ensemble, but the amplitude of the summer uptake in ORCHIDEE and SiBCASA is larger than that inferred by the inversions, and the peak of carbon emissions simulated by ORCHIDEE in August and September is also corrected by the inversions (respectively 0.04 and 0.07 PgC/month, compared to maximum ensemble posterior values of 0.02 and 0.04 PgC/month).

**Northern Europe**

In Northern Europe the range of posterior estimates is larger than that of the prior fluxes. All the simulations (including both prior and posterior) are well in phase, with a summer peak uptake in June/July and a stable winter flux between October and March. The size of the summer uptake varies by a factor three, between the -0.04 PgC/month as estimated by NAME in June and a corresponding value of -0.13 Pgc/month estimated by the FLEXINVERT+ inversion. The prior and posterior median are however nearly identical. Three inversions (CarboScope-Regional, FLEXINVERT+ and LUMIA) yield a clear annual net carbon sink (-0.09 to -0.14 PgC/year) for this region, however, the agreement on the size of the annual budget by CarboScope-Regional and FLEXINVERT+ is again by coincidence, as they distribute the fluxes very differently throughout the year.

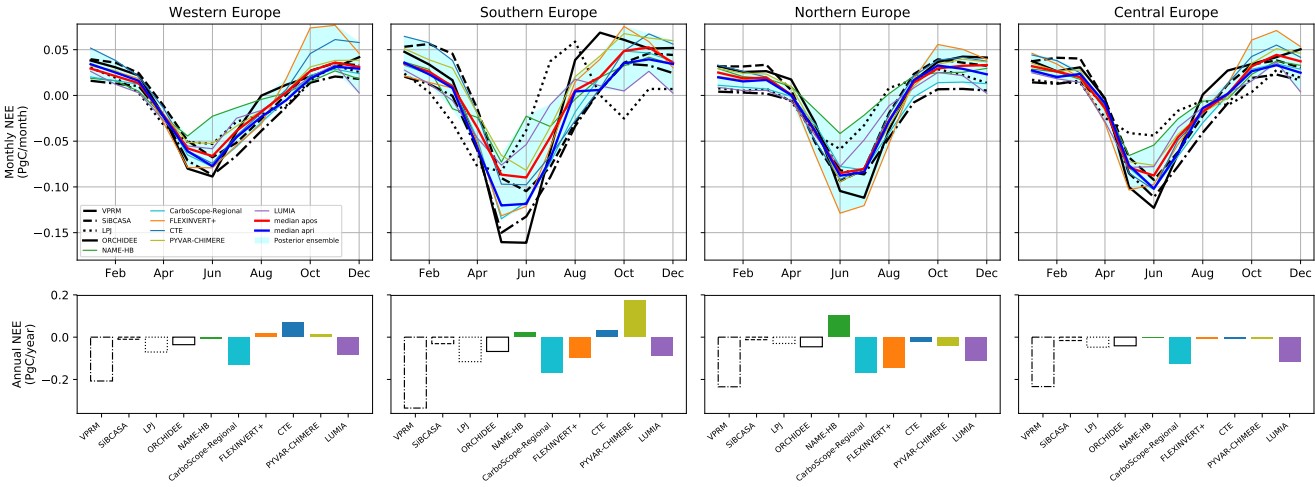

**Figure 9.** Upper row: Mean prior (black lines) and posterior (coloured lines) seasonal cycle of the terrestrial carbon flux in the four regions highlighted in Figure 1. Lower row: mean annual net terrestrial carbon flux for these same regions





## 5   Discussion

### 5.1   How well can regional-scale inversions constrain the annual budget of European NEE?

The annual budget of NEE is a key metric to characterise the amount of carbon absorbed by the European ecosystems, since it balances the releases in winter and at night (by ecosystem respiration) with the uptakes during daytime, mostly in spring and summer (by photosynthesis). Annual to multi-annual budgets are an important measure to quantify the impact of environmental conditions such as ecosystem management, disturbances and climate extremes on the terrestrial carbon cycle.

The annual budget has notably been synthesised in Reuter et al. (2017): on the one hand, global inversions that assimilate
only surface observations showed the geographical Europe as a moderate to rather small carbon sink ($\approx$-0.4 PgC/year) on multi-annual scales; and that, on the other hand, inversions constrained by satellite retrievals of total column atmospheric $CO_2$ ($XCO_2$) consistently infer that it is a much larger sink, on the order of -1 PgC/year. More recent studies suggest a smaller uptake: Scholze et al. (2019) find a mean sink of -0.3 $\pm$ 0.08 PgC/year by assimilating three datasets (namely in-situ atmospheric $CO_2$ and remotely sensed soil moisture and vegetation optical depth) into their Carbon Cycle Data Assimilation System. Similarly,
Crowell et al. (2019) estimate a mean sink of -0.25 $\pm$ 0.46 PgC/year from the ensemble of an intercomparison of atmospheric inversions based on $XCO_2$ observations from OCO-2. These estimates correspond to a geographical European domain which extends eastwards to the Ural, and which is much larger than the domain studied here. The areas of highest uptake in these satellite inversions are located in the eastern part of Europe, i.e. east of our EUROCOM domain.

From our ensemble of inversions we find a median sink of -0.21 PgC/year, relatively constant from year to year and with no
significant trend over the ten years of the period studied. Our study therefore tends to support the hypothesis that ecosystems in the European domain studied here are a weak carbon sink. Because of the differences in the domain extent, our inversions cannot close the controversy. But they indicate that, if there is a strong land sink over Europe (on the order of 1 PgC/year), then most of it has to be located in Eastern Europe, beyond the extent of our dense observation network.

Figure 10 provides results for our European domain (long term mean and IAV) from a set of state of the art global inversions
that assimilate only surface observations and which cover the time period studied here. They correspond to the set of global inversions used for the Global Carbon Project annual analyses (Le Quéré et al., 2018). The range of mean annual NEE obtained from these global inversions is about half that obtained from our regional inversions (0.8 PgC/year), which suggest that, at this scale, our regional inversions do not constrain the annual NEE better than global inversions. The spread between these 4 state-of-the-art global inversions selected for the GCP synthesis actually corresponds to the outcome of a long process of
improvement and selection of inversion configurations, as reflected by the very large spread of 1.8 PgC/year obtained from the inter-comparisons by Peylin et al. (2013). Therefore, one can expect the process of inter-comparing regional scale inversions started here with the EUROCOM project to yield a much-refined estimate of the annual to multi-annual budgets in the coming years. We note here again, that our inversion protocol was intentionally very loose, to allow for more systems to participate and hence to maximise the exploration of the space of uncertainties. It is therefore expected that the range of estimates would
be large, and we consider it as a rather conservative representation of the true uncertainties.





The slightly narrower spread of the global inversions nonetheless questions specific aspects of the regional inversions, which may prevent them from providing more precise estimates of the continental-scale fluxes. Part of the constraint on the European NEE in global inversions comes from the observed large scale atmospheric gradients between stations located in the Atlantic Ocean and Asia. These constraint are only incompletely transferred to regional inversion via their boundary conditions and the shorter scale gradients captured by the continental may not be sufficient to characterise the continental carbon balance. Unless the surface network is extended to cover sufficiently the Eastern and Southern parts of the domain, it might be useful to impose an constraint on the large scale gradients to the regional inversions. However this also mean that the relevant scale for regional inversions is possibly much smaller. The next section focuses thus on the spatial and temporal scales where our ensemble of inversions leads to robust and consistent results.

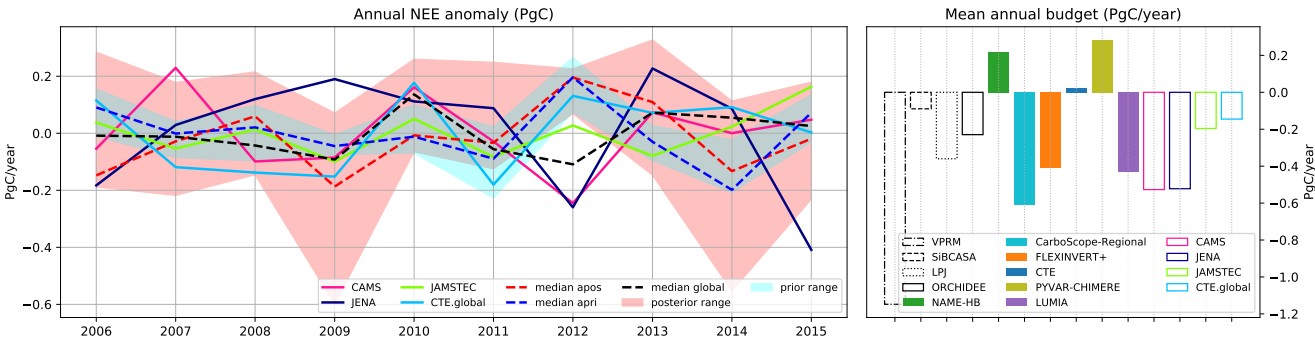

**Figure 10.** Comparison of the EUROCOM inversion ensemble with global inversions from the Global Carbon Project (Le Quéré et al., 2018). The right hand plot shows the annual NEE anomalies, with our prior and posterior ensembles shown as shaded areas, for the clarity of the figure. The right hand plot shows the mean annual budgets. The results from the global inversions were extracted for the exact EUROCOM domain on the global carbon atlas (www.globalcarbonatlas.org).

## 5.2 New insights on the European land carbon flux

While the net annual carbon flux is an important metric, focusing on it can give an overly pessimistic view of the results, especially integrated over the whole European domain for which the observational coverage is not homogeneous. In fact, a large share of the discrepancies between the inversions can be attributed to regions located at the eastern and southern borders of our domain that are not part of the four regions discussed in Section 4.2.3 (Russia, Ukraine, Turkey and North Africa, but also large swathes of the "Southern Europe" region).

Indeed, the ensemble of inversions leads to a narrower range of estimates than the ensemble of priors for regions with a dense network such as Central Europe (0.21 PgC/year of difference between the priors, vs. 0.13 PgC/year between the posterior estimates). In contrast, the range of the ensemble of the inversions is almost three times the range of the prior estimates (0.33 PgC/year between the optimised annual NEE, vs. 0.12 PgC/year between the prior estimates) in the parts of the domain that are outside the our four regions (see above), despite these regions being only rarely downwind of the observation network.This





means that although the posterior annual estimates at the continental (whole domain) scale may not be more robust than in e.g. the GCP global inversions (as discussed in 5.1), the regional inversions in our intercomparison are capable of resolving annual fluxes at the scale of large countries (e.g. 0.8 to 1.6 million km2 as for our four regions), provided that the observational network is dense enough.

The size of the spread between the posterior monthly flux estimates varies by a factor five throughout the year (at the continental scale). Monthly fluxes are usually well resolved in the first nine months of the year except for June, but this larger spread in June is due to one single inversion (NAME-HB), which among other differences uses a much reduced observation network (See Section 4.1) and covers only the last five years of our simulation period. The larger range among the estimates at the end of the year is more problematic and points to a problem of the inversions to robustly resolve the winter NEE, i.e .the

terrestrial respiration flux. Some speculative explanations could be larger systematic transport errors (winter concentrations are more difficult to represent, in particular at the highest latitudes, where the boundary layer remains extremely shallow and the vertical atmospheric stratification is high when the days are very short), and/or larger relative differences in the prescribed prior flux uncertainties between the inversions (uncertainties are overall smaller in winter because of the lack of photosynthesis).

    In three regions (Western, Central and Northern Europe), the prior and posterior ensemble median of the seasonal cycle are

almost identical, meaning that the inversions mostly provide a confirmation of the prior knowledge (see Figure 9). However, the differences between our ensemble median (i.e. best-informed guess) and each individual prior are sometimes large. For instance, the inversions consistently yield a summer uptake twice larger in Central Europe compared to the one computed by LPJ-GUESS. The results present therefore useful information for bottom-up modellers as they can be used to identify local or regional shortcomings in their models. This is also true when looking at the inversion results for Southern Europe. Although

the posterior estimates are not as consistent with each other as in the other regions, we nevertheless can clearly identify some shortcomings in the priors because they are far out of the ensemble spread (e.g. summer uptake by ORCHIDEE and SiBCASA, the double peak from LPJ-GUESS).

    In summary, the relative lack of convergence of the annual fluxes at the scale of the continent hides more robust features at monthly and smaller regional scales, especially in Central and Western Europe, where the observation network is the densest.

The divergences between the inversions regarding the winter fluxes will need to be investigated through a targeted effort. Nonetheless, the aim of optimising fluxes at country scale appears achievable for the large countries in areas with a dense network. In the later years, the density of observation sites in Northern Europe has increased a lot, so it is expected that the spread between the posterior estimates in this region can be reduced significantly in the future.

## 6   Conclusions and future of the EUROCOM project

The EUROCOM project delivered a set of NEE estimates at high temporal and spatial resolution at the disposal to the scientific community (Monteil et al., 2019). The data can be used as comparison and validation dataset for both bottom-up and inverse modellers. The input datasets (observations and prior fluxes) remain available for inverse modelling groups willing to submit





additional inversions, and we expect the size and robustness of the ensemble to grow over time. An extension of the inversions until 2019 is currently ongoing.

Our best posterior estimate (ensemble median) of the long-term mean annual terrestrial European NEE of -0.21 PgC/year over the years 2006-2015 is comparable to the median value of -0.3 PgC/year from our prior estimates as well as recent estimates from other studies (e.g. Scholze et al., 2019; Crowell et al., 2019), albeit for a slightly different domain). Since our domain here does not cover the European part of Russia, the area that is postulated to contribute most to the large European carbon sink (see e.g. Reuter et al., 2017)), we cannot resolve this controversy here with our intercomparison.

We deliberately kept the requirements in the intercomparison protocol (i.e. use of prescribed common data sets or inversion set ups) to a minimum (namely, prescribed fossil fuel emissions and common domain) to encourage the participation of voluntary contributions from regional atmospheric inverse modelling groups in this EUROCOM intercomparison project. Such an intercomparison approach, where a large number of parameters influencing results of the inversions vary from one system to another, presents the advantage that the resulting distribution of results provides a good approximation of the distribution of

uncertainties on the net European terrestrial carbon flux. Indeed, the analysis shows that no inversion is clearly more or less valid than the others and depending on the focus metrics, each can be an outlier. Such a multi-model/multi-inversion system ensemble is the best approach for providing robust estimates of the European carbon budget.

    The robustly modelled features in our ensemble are mainly the IAV and the mean seasonality of the annual $CO_2$ sink in regions with a dense observational network, i.e. mainly central and western Europe illustrating the usefulness of a coordinated

infrastructure such as ICOS in delivering high-quality observations. The coverage of the observational network in some regions of Europe is still limited, which is clearly reflected in a larger spread in the annual and monthly budgets in these regions within our ensemble. Observations from satellites, such as OCO-2 or the upcoming CO2M, may help in increasing the coverage but they have their own limitations (prone to clouds and aerosols, limited coverage during the winter season if based on passive optical instruments).

The mean annual terrestrial NEE itself is not strongly constrained by the observations and we find a spread of 0.8 PgC/year within our ensemble. As mentioned above, this is partly because of the high freedom in the choice of settings. This freedom in the choice of settings makes it rather challenging to fully understand the causes of the spread in the ensemble results and the underlying uncertainties. We will investigate these differences in more detail and evaluate some of the specific parameters involved in the inversion set-ups in a forthcoming paper. Eventually, this will lead to a much better quality of the regional

inversion estimates that could not have been possible without such an intercomparison exercise.

    Currently, the main benefit of regional inversions over global ones does not appear to be at the scale of the continent, but rather at finer spatial scales, in regions well covered by the observation networks. The observation network seems sufficiently dense to envision robust country-scale estimates of the carbon balance (at least for the largest countries) in Western and Central Europe. Recent expansions of the networks both in Northern and Southern Europe should also enable a significant reduction

of the spread between the inversions in the near future.



*Author contributions.* The intercomparison was collectively designed in the frame of the EUROCOM project coordinated by GB and MS with support from UK, GM, ML and PP. GM wrote the paper together with GB and MS. UK coordinated the construction of the common atmospheric observation database together with JT who compiled the pre-ICOS observations), processed the anthropogenic emission dataset, coordinated the exchange of observations, prior datasets and inversion results through the ICOS Carbon Portal, and supported the analysis of the results. GM designed and performed the LUMIA inversions. ML performed the PYVAR-CHIMERE inversions. CR provided the ocean flux used in some of the inversions, and together with CG designed the CarboScope-Regional system. FTK performed the CarboScope-Regional inversions. RLT developed the FLEXINVERT system and performed the FLEXINVERT inversions. EDW, AJM, EMW performed NAME model runs and processed NAME output for the high time frequency NAME-HB inversion. EDW, ALG and MR developed the NAME-HB inverse modelling method and code for performing high time frequency inversions and adapted the approach for the EUROCOM domain. ITvdLL and WP designed the CTE inversion system; ITvdLL and NES performed the CTE inversions and produced the SiBCASA prior. MM, PP and CG produced respectively the LPJ-GUESS, ORCHIDEE and VPRM priors.

*Competing interests.* The authors declare that they have no conflict of interest.

*Acknowledgements.* We thank the Swedish Research Council for funding the EUROCOM project (Dnr 349-2014-6576). MS and GM acknowledge support from the three Swedish strategic research areas ModElling the Regional and Global earth system (MERGE), the e-science collaboration (eSSENCE), and Biodiversity and Ecosystems in a Changing Climate (BECC). GM thanks the NSC at Lindköping University (part of the Swedish National Infrastructure for Computing, SNIC) for providing computer resources for the LUMIA simulations. GB and JT acknowledge the support by L. Rivier (for the transmission of pre-ICOS data), F. Chevallier and I. Pison (for the continuous development and maintenance of the PYVAR-CHIMERE system). FTK, CR, and CG thank the Deutsches Klimarechenzentrum (DKRZ) for use of the high-performance computing facilities. NES and ITvdLL benefited from computing resources from Netherlands Organization for Scientific Research (NWO; grant no. SH-312, 17616). NES was funded by NWO/OCW for Ruisdael/ICOS-NL. ITvdLL was funded by a NWO Veni grant (016.Veni.171.095). EW was funded by the UK Natural Environment Research Council (NERC) GW4+ Doctoral Training Partnership. ALG is funded through a NERC Independent Research Fellowship NE/L010992/1. MR was supported by NERC grants NE/K002236/1 and NE/S004211/1. RLT was funded by the Research Council of Norway through the research infrastructure project "Integrated Carbon Observation System (ICOS) Norway (grant no. 245927)". We thank Greet Janssens-Meanhout (European Commission, Joint Research Center, Ispra, Italy) for providing the fuel type and category specific version of the EDGAR v4.3 anthropogenic emission data and Hugo Denier van der Gon (TNO, The Netherlands) for making available the temporal emission profiles. We thank ICOS-ERIC for financial support. We also thank the data providers from the GLOBALVIEW-plus v3.2 product and from the WDCGG for the atmospheric $CO_2$ observations used in the inversions. Finally, we thanl the ICOS Atmospheric Thematic Center for providing the $CO_2$ measurements from CarboEurope-IP, GHG-Europe, and ICOS Preparatory Phase.



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
