# Peer review of "The regional EUROpean atmospheric transport inversion COMparison, EUROCOM: first results on European wide terrestrial carbon fluxes for the period 2006-2015"

_Atmospheric Chemistry and Physics, 2019_

## Short Comment (SC1) · 19 Dec 2019

I believe this study is very relevant for assessing the consistency of regional scale NEE estimates from regional inverse modeling systems.

I have some comments regarding the display of information:

- In figure 1, it is difficult to obtain information on the temporal density and continuity of measurements from the size of the dots and there is no key. An additional figure similar to figure 2 in Kountouris et al. (2018) or figure 2 in Rödenbeck et al.

[Figure]

(2003) would be more useful.

- Figure 2 and 3 would be more useful if it also included the sub-continental regions (at least as supplementary information).

- Figure 6 could be extend to also include other metrics, specifically correlations and standard deviation. I believe it would be more useful not to divide the plots by model but by metric in order to facilitate the model comparison. Order of the sites on the x-axis could be by latitude, longitude or altitude to observe if there are gradients.

It was my impression that since we are optimizing towards real data we are missing an assessment on how realistic the fluxes really are. Here I believe comparison of grid scale fluxes to Eddy Covariance measurements would be useful as well as comparison of spatial patterns (e.g. the spatial correlation and gradients) with satellite vegetation fluorescence products.

Furthermore, the use of dense measurement networks has the aim to distinguish small scale flux patterns. However, most of the analyses were at continental scale. I believe more analyses and discussion of the fluxes at the sub-continental scales is needed both for seasonal and interannual variability. At the interannual time scale, it would be useful to know if the variability shown by the inverse models reflects heat waves, droughts, cold spells, etc. If they are able to detect land use change or errors in the anthropogenic emissions.

Finally, the study only recommends increasing the density of the observation network, particularly in Southern and Eastern Europe. However, no analyses were made on the effects of the modeler's choices, e.g. measurement and prior errors, data selection, use of ocean fluxes. This choices could provide further recommendation for the development of these regional inverse modeling systems.

References:

Kountouris, P., Gerbig, C., Rödenbeck, C., Karstens, U., Koch, T. F., Heimann, M. (2018). Technical Note: Atmospheric $CO_2$ inversions on the mesoscale using data-driven prior uncertainties: methodology and system evaluation. Atmospheric Chemistry and Physics, 18(4), 3027–3045. http://doi.org/10.1029/2006JD008371

Rödenbeck, C., Houweling, S., Gloor, M., Heimann, M. (2003). $CO_2$ flux history 1982–2001 inferred from atmospheric data using a global inversion of atmospheric transport. Atmospheric Chemistry and Physics, 3, 2003.

---

## Referee Comment (RC1) · Anonymous Referee #1 · 27 Jan 2020

**Review of Monteil et al.: The regional EUROpean atmospheric transport inversion COMparison, EUROCOM: first results on European wide terrestrial carbon fluxes for the period 2006-2015**

General comments: the authors presented inverse biogenic fluxes estimates in Europe for over ten years (2006-2015) using the regional inversion technique as opposed to global inversions. These flux estimates were done under the protocol of using the same fossil fuel emission and a common database of in-situ $CO_2$ observations, while the transport models, inversion approaches, the choices of observation and prior biogenic fluxes differed. The authors concluded that at the continental scale, the European ecosystems are a relatively small sink (-0.21±0.2 PgC/year), consistent with the results demonstrated by the global inversions in the previous studies. This conclusion is quite out of my expectation and quite different than my experience. I have a few comments below that may potentially change the conclusions and maybe improve the results.

The manuscript is generally well written. The analysis is well presented. The authors provided a reasonable interpretation of their results and are definitely aware of the caveats of the study.

However, after reviewing the submission guidelines of ACP and GMD. I found this MS is a better fit for a GMD publication.  I am listing what I found below for the authors' reference.

> ACP research articles:
> "…Research articles must include substantial advances and general implications for the scientific understanding of atmospheric chemistry and physics. Manuscripts that report substantial new measurement results, but where the implications for atmospheric chemistry and physics are less developed, may be considered for publication as measurement reports (see below)…."

> GMD aims and scope:
> "…model experiment descriptions, including experimental details and project protocols;…"

This work has collected the inversion results from different groups without a well-designed protocol. It is very challenging to fully understand the cause of the spread of the ensemble and the underlying uncertainties, which limits the scientific advances of this work. The results overall match the previous studies which were drawn from the global inversions. There are a few technical aspects that could be improved, and I will list them as follows. I have quite a lot of faith that the author should deliver better inverse estimates if those technical improvements are implemented.

Given that it is an important paper for the EUROCOM project and can potentially provide some insight for future experiments, it is worth a publication. However, it's not clear to me how the authors handle boundary conditions and the related uncertainty in the inversions. The authors should have better clarifications on this aspect before re-submission.

For an ACP publication, however, I would recommend the following improvements to start with.

The authors built the results on top of a set of ensemble inversion results. The only protocol, for now, is to the same fossil fuel emission and a common database of in-situ $CO_2$ observations. To understand what caused the large spread of the ensemble, fixing at least one of the model components is required. That's said, the authors should at least have one set of results using the same transport model, same prior fluxes, same boundary conditions, or the same observations. without the common setup, it limits the depth of the scientific understanding of this work.

The regional inverse results in the MS do not appear to have better constraints than the global inversions. Technically, regional inversions can be driven by mesoscale transport. All of the experiments in the MS were driven with reanalysis or forecast data at ~101 to 102 km, and most of the meteorological forcing data do not have TKE. I suspect that these limitations are the main reasons that lead to unexpected equivalent results to the global inversions. I strongly recommend the working groups to use the mesoscale model output as the meteorological forcing for future experiments.

As I mentioned before, the boundary conditions need to be stated in more detail.

The authors mentioned that the locations of some observations are very challenging for transport models in section 4.1. It may be a good idea to remove those challenging sites before inversions to avoid large transport errors. A detailed model-data mismatch would be more appreciated and help to improve the inverse flux estimates.

Specific comments:

1.     Line 25, spell "NBP" out.
2.     Line 187, remove of "full"
3.     Figure 2, do the authors know why VPRM looks so different than others?
4.     Table 2, the author can one row of the choice of observation for each group.
5.     Line 485, I would be not surprised by the smaller error reduction of the annual results than that of the monthly results due to the annual net NEE is close to zero.
6.     Line 704, change km2 to $km^2$

---

## Referee Comment (RC2) · Anonymous Referee #2 · 1 Apr 2020

This paper presents the first results of the EuroCom project, with an inter-comparison of net terrestrial ecosystem exchange estimated by 6 regional inversion systems following a flexible protocol that allows maximum participation and sensitivities to different transport, priors and number of in situ observations assimilated.

General comments

The manuscript is well written and the explanations are clear overall. The paper would benefit by explaining in more detail what is the purpose of comparing such wide range

of diverse systems which produce such large spread in the optimized fluxes. What do we learn from such exercise? It seems that one of the messages is that all these differences in the configuration of the inversion systems have a large influence on the resulting optimized fluxes, i.e. assumptions in prior uncertainty estimation and data assimilation methodology, transport model, temporal/spatial resolution, boundary conditions, number of observations used, ocean fluxes, etc. The results point that regional inversions using the currently available in situ data in Europe are not able to properly constrain the NEE at regional scale, and the spread between different optimized fluxes is as large as the mean or median flux. The other aspect that could be improved is the presentation of the uncertainty from each individual inversion system. I have not seen the posterior uncertainty in any of the plots. It would be useful to add this information in the bar plots where the estimate from each system is compared.

Specific comments/questions:

1. The ICOS network is currently not a high density in-situ surface observation network (with 19 stations run by 12 countries as described in line 64).

2. The paper only addresses large regional budgets at subcontinental scale, not country scale budgets. Why not look at budgets for a relatively large country where there are enough observations, e.g. France or Germany to demonstrate the capability at country scale?

3. The use of mesoscale transport model is not appropriate as mesoscale weather systems have scales of less than 100 km you need higher resolution than 10 to 100 km to resolve them. It would be best to replace mesoscale model with regional model.

4. What are the implications of not correcting for errors in the anthropogenic emissions and ocean fluxes? The signal of the anthropogenic emissions vary during the day. So if inversions use observations at slightly different times of day, the influence of the anthropogenic emission error on the optimized flux will also vary. Could this explain part of the divergence between the optimized fluxes from the different inversion systems?

Minor comments:

-Line 26: Define NBP

-Line56: …that does not smooth …

-Line 119: Replace "the find" by "finding".

-Line 124: Please define "model error", "representation error" and "aggregation error".

-Line 186: Remove the extra "full".

-Lines 193-203: Temporal resolution of ORCHIDEE prior is missing.

-Lines 193-209: Spatial resolution of the prior from ORCHIDEE and PJ-GUESS is missing.

-Line 228: PgC/months? Shouldn't the units be PgC/month?

-Line 233: Please include resolution of EDGARv4.3 inventory

-Lines 241-243: Biomass burning emissions can be large over the summer in the Mediterranean region over the summer (e.g. 2007 and 2015). Could this also explain part of the large divergence between optimized fluxes in that region?

-Lines 253-257: The Takahashi et al. (2009) climatology will underestimate the ocean sink for the period 2006-2015, so this will explain part of the discrepancy between NAME-HB and other inversion systems. Other relatively out-of-date ocean data sets might lead to also an underestimation of the ocean sink. Does this mean that in those systems that do not correct the ocean fluxes, the error in the ocean sink will be attributed to NEE?

-Line 304: What is the Rödenbeck approach?

-Line405: Remove extra bracket after Radon.

-Line 409-411: What about the representation error associated with resolution of transport model? Shouldn't it be part of the observation uncertainty? Representation errors tend to be very large at sites close to anthropogenic emissions.

-Line 416: How much does the observation uncertainty vary from site to site?

-Line 427: Is FLEXINVERT+ the only inversion system in which the uncertainties in the fossil fuel emission estimates contribute to the observation error? It seems to be this is an important uncertainty to consider given that most in situ sites in Europe are affected by anthropogenic emissions.

-Line 479: "if" should be "it".

-Line 486: "diagnostics" should be "diagnostic".

-Figure 6: The differences between the prior and posterior appear to be very small at most sites. Does this mean that the prescription of the prior uncertainty is too small and transport uncertainty too large?

-Lines 600-603: The three sentences starting with "The figure..." would fit better in the caption of Figure 9.

-Line 608: The Central European NEE is only robust in the sign of the budget, but not the magnitude (as shown in lower righ most panel in Figure 9).

-Figure 9: It is not possible to read the figure caption.

-Line 687: "an constraint" should be "a constraint".

-Line 700: "the our four regions" should be "the four regions".

-Lines 698-700: If there is a deterioration in the optimized fluxes with respect to the prior fluxes in data sparse regions, doesn't this mean that the assumptions in the inversion are not correct? One would expect that the optimized fluxes are always better or the same than the prior fluxes (where there are no observations).

-Line 730: The use of high resolution is relative. It's probably best if you specify the

range of spatial and temporal scales resolved by the regional inversion systems. A resolution of 0.5 degrees is not considered by most as high spatial resolution. Temporal resolution is not high either if observations are filtered in time to short afternoon and nighttime windows.

-Line 731: Given the spread of the optimized fluxes is so large, can the data be used as a validation data set?

-Lines 735-736: Please include the associated uncertainty of the posterior estimate.
* * *

---

## Author Comment (AC1) · 10 Jul 2020

**Response to review 1**

**General comments: the authors presented inverse biogenic fluxes estimates in Europe for over ten years (2006-2015) using the regional inversion technique as opposed to global inversions. These flux estimates were done under the protocol of using the same fossil fuel emission and a common database of in-situ CO2 observations, while the transport models, inversion approaches, the choices of observation and prior biogenic fluxes differed. The authors concluded that at the continental scale, the European ecosystems are a relatively small sink (-0.21±0.2 PgC/year), consistent with the results demonstrated by the global inversions in the previous studies. This conclusion is quite out of my expectation and quite different than my experience. I have a few comments below that may potentially change the conclusions and maybe improve the results.**

**The manuscript is generally well written. The analysis is well presented. The authors provided a reasonable interpretation of their results and are definitely aware of the caveats of the study.**

**However, after reviewing the submission guidelines of ACP and GMD. I found this MS is a better fit for a GMD publication. I am listing what I found below for the authors' reference.**

> **ACP research articles:**

> **"...Research articles must include substantial advances and general implications for the scientific understanding of atmospheric chemistry and physics. Manuscripts that report substantial new measurement results, but where the implications for atmospheric chemistry and physics are less developed, may be considered for publication as measurement reports (see below)...."**

> **GMD aims and scope:**

> **"...model experiment descriptions, including experimental details and project protocols;…"**

ACP is one of the reference journals for publishing inverse modelling studies and for inversion inter-comparisons (e.g. www.atmos-chem-phys.net/13/9039/2013/, www.atmos-chem-phys.net/18/3047/2018/, www.atmos-chem-phys.net/16/1289/2016/, www.atmos-chem-phys.net/15/12765/2015). Our paper is in line with these studies and helps assessing the robustness of their findings (beyond the sensitivity tests or the diagnostics that individual inverse modellers usually run themselves). We derive conclusions regarding the European NEE that are of general interest for the scientific community (such as the one raised above by the reviewer himself regarding the European mean sink), and even though our study domain is Europe, some of the conclusions are relevant for inversions focusing on other parts of the world (the same models are used in different regions). We therefore disagree with the reviewer's comment and think that ACP is a perfectly well suited target journal for our study.

**This work has collected the inversion results from different groups without a well-designed protocol. It is very challenging to fully understand the cause of the spread of the ensemble**

**and the underlying uncertainties, which limits the scientific advances of this work. The results overall match the previous studies which were drawn from the global inversions. There are a few technical aspects that could be improved, and I will list them as follows. I have quite a lot of faith that the author should deliver better inverse estimates if those technical improvements are implemented.**

The protocol was designed according to the objectives of the intercomparison. Here, it was not specifically to 'deliver better inverse estimates' (we rather consider it a task for each of the participating modeller to constantly improve their system and their estimates), but to provide a snapshot of the state of the art, and, more importantly, to explore the full range of uncertainties resulting out of an ensemble of sensible assumptions for the inversion set-up. ¶

Another objective for designing the protocol was to keep the extra work load for participating in the intercomparison low to allow for a large number of participants (only two groups were funded for this work).

Eventually the EUROCOM project has and will lead to improvements in the inversion systems, but that is not specifically the aim of this paper.

**Given that it is an important paper for the EUROCOM project and can potentially provide some insight for future experiments, it is worth a publication. However, it's not clear to me how the authors handle boundary conditions and the related uncertainty in the inversions. The authors should have better clarifications on this aspect before re-submission.**

The boundary conditions are specific to each system, and briefly described in Section 3.3.1 (which provides references for a more complete description for each system). We have reorganized the text of this section (and renamed it) to make it clearer.

It is not possible to use the same boundary condition in all systems since they rely on different types of transport models or configurations (for instance, CTE does not have boundaries since it is a global model, FLEXINVERT and LUMIA both rely on FLEXPART for regional transport, but FLEXINVERT uses global FLEXPART simulations while LUMIA uses regional ones, so essentially, the BC used in one inversion would not be relevant for the other one).¶

The uncertainty associated to the boundary condition (for the systems that have one) is generally considered part of the observational uncertainty (except in CTE (global) and NAME (which optimizes the BC as well)) and described in Section 3.3.3. Not all systems explicitly differentiate the uncertainty associated to the boundary condition from other uncertainties in the observation space (model error, observation error, etc.), as it is not easy to quantify them individually, but references are provided to papers describing the individual systems and justifying in more details these choices of uncertainties.

**For an ACP publication, however, I would recommend the following improvements to start with. The authors built the results on top of a set of ensemble inversion results. The only protocol, for now, is to the same fossil fuel emission and a common database of in-situ CO2 observations. To understand what caused the large spread of the ensemble, fixing at least one of the model components is required.**

Actually, to really understand what caused the large spread of the ensemble, an ensemble of ensemble would be needed, fixing, in each ensemble, one of the numerous parameters (prior

fluxes, parameters of the prior uncertainty, control resolution, etc.) of the inversions which are potential sources of uncertainty. ¶

**That's said, the authors should at least have one set of results using the same transport model, same prior fluxes, same boundary conditions, or the same observations. without the common setup, it limits the depth of the scientific understanding of this work.**

By experience, we know that many parameters are uncertain and yield a significant portion of the resulting uncertainty. Weighting each contribution is a very expensive (computationally) and long exercise requiring lots of analysis. This was clearly out of the scope and capabilities of this first analysis of the inter-comparison.

Furthermore, in practice, many of the key parameters driving the uncertainty (the transport model, the formulation of the control vector, etc.) cannot be imposed to all inversion systems. The observation selection, the treatment of the boundary conditions (see above) and the model errors should depend on the transport model etc.

Our choice of a very loose protocol makes it more difficult to systematically analyse the influence of specific components of the inversion systems, but the results provide a realistic range for the European NEE, which was one of the main objectives of the intercomparison.

We have edited the introduction of Section 3 to clarify the justifications for our protocol.

**The regional inverse results in the MS do not appear to have better constraints than the global inversions. Technically, regional inversions can be driven by mesoscale transport. All of the experiments in the MS were driven with reanalysis or forecast data at ~101 to 102 km, and most of the meteorological forcing data do not have TKE. I suspect that these limitations are the main reasons that lead to unexpected equivalent results to the global inversions. I strongly recommend the working groups to use the mesoscale model output as the meteorological forcing for future experiments.**

There were some mistakes in Table 2, which have been corrected in the revised manuscript and may have lead to some misunderstanding: several of the experiments were in fact driven by higher resolution meteorology (IFS operational forecast and UK Met-Office model).

That said, the equivalence of results with global inversions is not that unexpected: global inversions are supposed to derive accurate large-scale constraints. The advantage of regional inversions is their capacity to assimilate data from more sites, and derive constraints at higher resolution where and when the observation coverage is sufficient. As mentioned in the discussion, the regional inversions used here and the global inversions to which they are compared are also not at the same stage of maturity.

**As I mentioned before, the boundary conditions need to be stated in more detail.**

See our answer regarding this above.

**The authors mentioned that the locations of some observations are very challenging for transport models in section 4.1. It may be a good idea to remove those challenging sites before inversions to avoid large transport errors. A detailed model-data mismatch would be more appreciated and help to improve the inverse flux estimates.**

In future inversions, it will indeed be necessary for some systems to pay more careful attention to these challenging sites (especially HEI and GIF). This can be done by either excluding them or by applying a stricter data selection. Several systems account for this problem already by inflating the observation uncertainty, so that these sites do not have large impact on the results, but others, the impact of these sites can be strong, locally. This issue prevents us from analyzing the fluxes in details in the vicinity of these sites, but at the scale of the domain, or even of the larger regions used in the paper, other source of uncertainty dominate, and we are confident that this has only a minor impact on our conclusions at this stage.

It is difficult to propose much more detailed model-data mismatch than what is already in the paper and SI: there are too many inversions, too many sites and too many years to show everything, and the problematic sites or time of the day/year are not the same for all systems, so it is not even feasible to highlight specifically remarkable time series.

**Specific comments:**

**1. Line 25, spell "NBP" out.**

We have fixed this.

**2. Line 187, remove of "full"**

We have fixed this.

**3. Figure 2, do the authors know why VPRM looks so different than others?**

Yes, this is in fact already explained in the manuscript (end of Section 3.2.1). VPRM is a diagnostic model (it assimilates eddy-covariance flux observations using a very simplified biosphere model, that can lead to a near zero respiration in winter in large parts of the domain). The CarboScope Regional (CSR) inversion accounts for this by adjusting an annual bias correction in addition to the 3-hourly NEE (this is one example where a stricter protocol forcing some systems to use a prior for which they are not designed might further degrade the results, see above).

**4. Table 2, the author can one row of the choice of observation for each group.**

We have added a row in Table 2. Note that this is a simplified description, and does not include the site selection (for space limitations, and because it is easy to get it from Figure 6). A full overview of the data selection is provided in Figure SI2.

**5. Line 485, I would be not surprised by the smaller error reduction of the annual results than that of the monthly results due to the annual net NEE is close to zero.**

Indeed, but we are not saying that it is surprising, we are just stating it as a result here, and we believe it is worthwhile doing so.

**6. Line 704, change km2 to km 2**

We have fixed this.

**Response to review 2**

**This paper presents the first results of the EuroCom project, with an inter-comparison of net terrestrial ecosystem exchange estimated by 6 regional inversion systems following a flexible protocol that allows maximum participation and sensitivities to different transport, priors and number of in situ observations assimilated.**

**The manuscript is well written and the explanations are clear overall. The paper would benefit by explaining in more detail what is the purpose of comparing such wide range of diverse systems which produce such large spread in the optimized fluxes. What do we learn from such exercise? It seems that one of the messages is that all these differences in the configuration of the inversion systems have a large influence on the resulting optimized fluxes, i.e. assumptions in prior uncertainty estimation and data assimilation methodology, transport model, temporal/spatial resolution, boundary conditions, number of observations used, ocean fluxes, etc.**

As explained in our reply to reviewer #1, the main aim is, at this stage, to assess the range of uncertainties that comes out from regional inversions (beyond the typical sensitivity tests that individual modellers typically run). This requires an ensemble maximizing the variability of the inversion setups. On the contrary, identifying the specific contribution of individual settings/design choices in the inversions requires a very controlled protocol (essentially only varying that parameter we want to estimate the influence of). The two aims are therefore difficult to reconcile in a single study, and we focused on the first one.

We have clarified that aspect of our paper in the introduction, and have also explained better the motivations for our protocol in the introduction of Section 3.

**The results point that regional inversions using the currently available in situ data in Europe are not able to properly constrain the NEE at regional scale, and the spread between different optimized fluxes is as large as the mean or median flux.**

This comparison with the mean or median flux is not totally relevant: the net flux (NEE) is relatively small, but the gross fluxes (GPP and respiration) are very large, and the uncertainty that arises from our ensemble should rather be compared to the uncertainty on these terms. The inversions do manage to reduce the range of estimates compared to the priors (and more generally compared to bottom-up models) regarding key features such as the shape of the seasonal cycle. The lack of convergence regarding the annual budget and the IAV at the continental scale are also interesting findings in themselves, as it was not initially expected and will certainly motivate further developments and exchanges between the inverse modellers, that would not have happened if that exercise hadn't been performed.

**The other aspect that could be improved is the presentation of the uncertainty from each individual inversion system. I have not seen the posterior uncertainty in any of the plots. It would be useful to add this information in the bar plots where the estimate from each system is compared.**

It is actually difficult to obtain such a metric for all systems (see the lack of posterior uncertainties e.g. in Peylin et al., 2013, BGD). Not all of them can technically or practically compute it, and those who can usually do not compute it in ways that are easily comparable. Low

or reduced rank inversion approaches can access it through analytical compations but variational inversions must be coupled to complex minimization schemes or Monte Carlo experiments to produce an approximation of the theoretical posterior uncertainty (Kadygrov et al., 2015, ACP). The uncertainty reduction that is sometimes computed in inversions is a diagnostic that is useful in some contexts (e.g. for network design studies with OSSEs using a single inversion framework), but it is highly theoretical (it strongly relies in all the statistical assumptions made by the inversion system) and it could be misleading (Henne et al., 2016, ACP).

**Specific comments/questions:**

**1. The ICOS network is currently not a high density in-situ surface observation network (with 19 stations run by 12 countries as described in line 64).**

The "high density" qualifier is subjective, but ICOS is clearly one of the densest continental-scale networks available to date. We have replaced "high-density" by the slightly more neutral "dense".

**2. The paper only addresses large regional budgets at subcontinental scale, not country scale budgets. Why not look at budgets for a relatively large country where there are enough observations, e.g. France or Germany to demonstrate the capability at country scale?**

Although in some parts of Europe, the network might be dense enough for allowing some national-scale NEE estimation, this is rather the exception than the rule. Furthermore in those countries where it might be feasible, we do not think that our setups are the most appropriate: the resolution of our inversions (transport model, control vector, prior and other fluxes) is still coarse in contrast to the size of European countries. It might in fact be possible to obtain better or at least more consistent results at that country scale, using the same observational data but an experimental setup dedicated to that scale (smaller domain, higher-resolution fluxes, dedicated prior, etc.). The country scale is politically/societally sensitive therefore we prefer to remain on the cautious side (it should also be . The scale presented in the paper is the one at which we think our inversions are the most relevant.

**3. The use of mesoscale transport model is not appropriate as mesoscale weather systems have scales of less than 100 km you need higher resolution than 10 to 100 km to resolve them. It would be best to replace mesoscale model with regional model.**

As mentioned in our reply to reviewer #1, there were some errors in Table 2, listing the meteorological input data used by the different transport models. We have corrected this.

The resolution of the transport model is limited by the computational cost of high resolution transport simulations (the computational cost increases exponentially with the resolution). Furthermore, the spatial resolution of the model has to be adapted to that of the observation network. With observation sites separated by, at least a few hundreds km, our observation network does not actually provide very fine-scale constraints, so the benefit of using higher resolution transport model would be limited to a few very specific cases (sites with complex orography, or in vicinity of strong point sources).

**4. What are the implications of not correcting for errors in the anthropogenic emissions and ocean fluxes? The signal of the anthropogenic emissions vary during the day. So if inversions use observations at slightly different times of day, the influence of the**

**anthropogenic emission error on the optimized flux will also vary. Could this explain part of the divergence between the optimized fluxes from the different inversion systems?**

The impact of the ocean fluxes on the observations is very small, on the order of the observation uncertainty. It is unlikely that it can explain a significant part of the divergences.

The impact of anthropogenic emissions on the observations can definitively be important even though the signal at the stations used here is dominated by the NEE, which is why we all use the same anthropogenic emissions. Errors in these emissions could be affecting the inversions differently, but not just because they use observations at slightly different times, but also because all the transport models do not represent the sensitivity of the observation to the emissions with the exact same accuracy, and because not all inversions construct their observation error in the same way.

**Minor comments:**

**-Line 26: Define NBP**

Done

**-Line56: . . .that does not smooth . . .**

Done

**-Line 119: Replace "the find" by "finding".**

Done

**-Line 124: Please define "model error", "representation error" and "aggregation error".**

The model error is the error due to lack of accuracy of the transport model (i.e. the error it would make even if it was driven by the true fluxes). The representation error is due to the representation of discrete or fine-resolution processes (fluxes, observations) on a coarser model grid. The aggregation error accounts for errors due to the control of fluxes at a coarser resolution than that of the transport model.

We have slightly simplified the sentence to remove the aggregation error (which isn't formally accounted for by most of the inversions)

**-Line 186: Remove the extra "full".**

Done

**-Lines 193-203: Temporal resolution of ORCHIDEE prior is missing.**

The temporal resolution of the ORCHIDEE product is 3-hours. We have clarified this.

**-Lines 193-209: Spatial resolution of the prior from ORCHIDEE and PJ-GUESS is missing.**

The spatial resolution is 0.5°, we have clarified this.

**-Line 228: PgC/months? Shouldn't the units be PgC/month?**

Yes, we have corrected this.

**-Line 233: Please include resolution of EDGARv4.3 inventory**

The original resolution is 0.1°, but the product was regridded to 0.5° for the simulations. This has now been specified in the manuscript.

**-Lines 241-243: Biomass burning emissions can be large over the summer in the Mediterranean region over the summer (e.g. 2007 and 2015). Could this also explain part of the large divergence between optimized fluxes in that region?**

It may play a role, but given that the two systems that included a biomass burning product are at the two opposite extreme of the interval for summer emissions in Southern Europe, it is unlikely that this flux explains a large part of the differences. The Southern Europe region includes parts of the domain that are not well constrained by the observations (Greece and the Balkans), and on the other hand, there are many sites in Spain, which are not easy to represent. Furthermore, the divergences between the prior are the strongest in that part of our domain.

**-Lines 253-257: The Takahashi et al. (2009) climatology will underestimate the ocean sink for the period 2006-2015, so this will explain part of the discrepancy between NAME-HB and other inversion systems. Other relatively out-of-date ocean data sets might lead to also an underestimation of the ocean sink. Does this mean that in those systems that do not correct the ocean fluxes, the error in the ocean sink will be attributed to NEE?**¶

Yes, this is an inevitable feature of inversions: errors in non-optimized components of the inversion map into the optimized solution. It is however unlikely that the choice of ocean fluxes explain any significant fraction of the differences between the inversions: the impact of ocean fluxes (regardless of which ocean flux dataset is used) at the observation sites is very small and nearly negligible compared to that of the anthropogenic and biogenic fluxes (see figure below).

[Figure]

Figure 1: Mean absolute impact of the ocean (CarboScopev1.5), biosphere (LPJ-GUESS prior) and fossil flux estimates at the observation sites, in the LUMIA simulations. The error-bars mark the standard deviation over the 10 year simulation period.

**-Line 304: What is the Rödenbeck approach?**

In four of the systems (LUMIA, CHIMERE, CSR and NAME), the background concentration corresponds to the transport to the observation sites of a boundary condition at the edge of the domain. In the Rödenbeck approach, used in CSR and LUMIA, it is the global transport model

from which that boundary condition is extracted which is used to transport it to the observation sites (while in CHIMERE and NAME, this is done directly by the regional transport model). The approach is described and justified in details in (Rödenbeck et al., 2009), we do not think that it is relevant to re-explain it in our manuscript.

**-Line405: Remove extra bracket after Radon.**

Done

**-Line 409-411: What about the representation error associated with resolution of transport model? Shouldn't it be part of the observation uncertainty? Representation errors tend to be very large at sites close to anthropogenic emissions.**

Yes, it is part of the "uncertainty of the forward transport model". The paragraph has been clarified.

**-Line 416: How much does the observation uncertainty vary from site to site?**

That part of the manuscript was in fact incorrect. The average uncertainty is on the order of 2 ppm and a minimum uncertainty of 1 ppm has been enforced for each obs. The average uncertainty ranges from 1.02 ppm (Mace Head) to 4 ppm (Puijo) for the year 2015. But these mean values hide larger variations: the uncertainty only reaches a maximum of 2.55 ppm at Lampedusa while it goes up to 32.7 ppm at Cabauw. The manuscript has been corrected.

**-Line 427: Is FLEXINVERT+ the only inversion system in which the uncertainties in the fossil fuel emission estimates contribute to the observation error? It seems to be this is an important uncertainty to consider given that most in situ sites in Europe are affected by anthropogenic emissions.**

It is the only system to account for it explicitly, but the way other systems account for it when constructing their observation uncertainty vector. The uncertainties in FLEXINVERT are actually in the lower range of the ensemble.

**-Line 479: "if" should be "it".**

Done

**-Line 486: "diagnostics" should be "diagnostic".**

Done

**-Figure 6: The differences between the prior and posterior appear to be very small at most sites. Does this mean that the prescription of the prior uncertainty is too small and transport uncertainty too large?**

Not really. Larger prior uncertainties (or lower observation error) would lead to a better posterior fit to the observations, but not necessarily to more realistic flux adjustments (the inversion systems can "over-fit" the data, i.e. adjust fluxes to compensate the various model or observation errors). The balance of prior and observation uncertainties are set by the modellers in accordance to their experience with their inversion systems.

**-Lines 600-603: The three sentences starting with "The figure. . ." would fit better in the caption of Figure 9.**

Indeed, we have fixed this.

**-Line 608: The Central European NEE is only robust in the sign of the budget, but not the magnitude (as shown in lower righ most panel in Figure 9).**

The "most robust" qualifier is given in comparison to the other regions.

**-Figure 9: It is not possible to read the figure caption.**

We have edited the figure to improve the readability, and repeated part of the information in the caption.

**-Line 687: "an constraint" should be "a constraint".**

Done

**-Line 700: "the our four regions" should be "the four regions".**

Done

**-Lines 698-700: If there is a deterioration in the optimized fluxes with respect to the prior fluxes in data sparse regions, doesn't this mean that the assumptions in the inversion are not correct? One would expect that the optimized fluxes are always better or the same than the prior fluxes (where there are no observations).**

This expectation would be verified if the inversion were all constrained with perfectly adequate uncertainty statistics (observations and prior uncertainties), which cannot be the case. In data sparse areas, the inversion strongly relies on the prior error covariance matrices to extrapolate the information from the few available observations. If these covariances poorly match the actual spatial distribution of NEE, the extrapolation can be highly erroneous. In regions where there are a lot of data, the NEE is more directly constrained by the data, and therefore its estimation is more robust.

**-Line 730: The use of high resolution is relative. It's probably best if you specify therange of spatial and temporal scales resolved by the regional inversion systems. A resolution of 0.5 degrees is not considered by most as high spatial resolution. Temporal resolution is not high either if observations are filtered in time to short afternoon and nighttime windows.**

The high spatial and temporal resolution is indeed relative to the atmospheric inversion field (global inversions run at best at a 2° resolution). But we have modified the lines according to the reviewer's suggestion

**-Line 731: Given the spread of the optimized fluxes is so large, can the data be used as a validation data set?**

The spread of bottom-up models is actually even larger. While none of the inversions should be used individually as a validation dataset, the ensemble statistics (mean/median and spread/standard-deviation provide a good representation of the likely interval of NEE that can be derived from atmospheric $CO_2$ observations. We have nonetheless removed the sentence as it indeed can be confusing.

**-Lines 735-736: Please include the associated uncertainty of the posterior estimate.**

Done

**Response to short comment 1**

**I believe this study is very relevant for assessing the consistency of regional scale NEE estimates from regional inverse modeling systems.**

**I have some comments regarding the display of information:**

**In figure 1, it is difficult to obtain information on the temporal density and continuity of measurements from the size of the dots and there is no key. An additional figure similar to figure 2 in Kountouris et al. (2018) or figure 2 in Rödenbeck et al. (2003) would be more useful.**

The main purpose of the figure is to show the location of the observation sites and the definition of the regions used further in the paper. The size of the dots is really only qualitative, as the amount of constraint each site provides depends on the specific data selection and uncertainty attribution in each system.

**Figure 2 and 3 would be more useful if it also included the sub-continental regions (at least as supplementary information).**

We now provide regional versions of Figure 2 and 3 in supplementary information

**Figure 6 could be extend to also include other metrics, specifically correlations and standard deviation. I believe it would be more useful not to divide the plots by model but by metric in order to facilitate the model comparison. Order of the sites on the x-axis could be by latitude, longitude or altitude to observe if there are gradients.**

We have tried various presentations of the model-data mismatches and settled on that one as it suited best our needs (presenting a synthetic overview of the model-data mismatches).

**It was my impression that since we are optimizing towards real data we are missing an assessment on how realistic the fluxes really are. Here I believe comparison of grid scale fluxes to Eddy Covariance measurements would be useful as well as comparison of spatial patterns (e.g. the spatial correlation and gradients) with satellite vegetation fluorescence products.**

In theory, we agree that the comparison with EC data would be useful, but it is not as simple as just comparing the EC and inverse-modelling derived fluxes: they are representative of very different spatial scales (a few km2 for EC data, a few thousands of km2 for inversions), therefore either the inversions need to be subsampled or the EC data needs to be upscaled, neither of which is straightforward. Furthermore, EC fluxes come with their own uncertainties and biases. Doing the comparison properly could practically be a study on its own. Likewise, the comparison with fluorescence data and inversion fluxes should only be done in a dedicated study (fluorescence is a proxy of GPP, not of NEE, the direct comparison between the two is not possible).

**Furthermore, the use of dense measurement networks has the aim to distinguish small scale flux patterns. However, most of the analyses were at continental scale. I believe more**

**analyses and discussion of the fluxes at the sub-continental scales is needed both for seasonal and interannual variability. At the interannual time scale, it would be useful to know if the variability shown by the inverse models reflects heat waves, droughts, cold spells, etc. If they are able to detect land use change or errors in the anthropogenic emissions.**

Here again, we agree that some of these analyses could be interesting, but we do not think that they should be part of this paper. Our inversions span a period of 10 years and the whole European continent. The list of potential climatic anomalies that could be studied is endless, and the paper is already long. This type of analysis is better done in papers dedicated to studying these specific climatic events, and crossing more information sources than just inversions. Our study can then help assessing the relevance of inversions in that context.

**Finally, the study only recommends increasing the density of the observation network, particularly in Southern and Eastern Europe. However, no analyses were made on the effects of the modeler's choices, e.g. measurement and prior errors, data selection, use of ocean fluxes. This choices could provide further recommendation for the development of these regional inverse modeling systems.**

We have partly replied to this in our answers to the anonymous reviewers: the goal of this study is to identify and quantify discrepancies between the inversion results (and therefore estimate the robustness of regional inversions). Identifying the contribution of different modeler's choices to that overall uncertainty is definitely one of the aims of the EUROCOM project, but it is outside the scope of this paper (it is difficult to do both: estimating the overall uncertainty calls for an ensemble as diverse as possible, while estimating the contribution of one specific choice requires a more controlled protocol). We have clarified the scope of this paper in the introduction and explained better our experiment design in Section 3.

**References**

Peylin, P., Law, R. M., Gurney, K. R., Chevallier, F., Jacobson, A. R., Maki, T., Niwa, Y., Patra, P. K., Peters, W., Rayner, P. J., Rödenbeck, C., Van Der Laan-Luijkx, I. T., & Zhang, X. (2013). Global atmospheric carbon budget: Results from an ensemble of atmospheric CO2 inversions. Biogeosciences, 10(10), 6699–6720. https://doi.org/10.5194/bg-10-6699-2013

Henne, S., Brunner, D., Oney, B., Leuenberger, M., Eugster, W., Bamberger, I., Meinhardt, F., Steinbacher, M., & Emmenegger, L. (2016). Validation of the Swiss methane emission inventory by atmospheric observations and inverse modelling. Atmospheric Chemistry and Physics, 16(6), 3683–3710. https://doi.org/10.5194/acp-16-3683-2016

Kadygrov, N., Broquet, G., Chevallier, F., Rivier, L., Gerbig, C., & Ciais, P. (2015). On the potential of the ICOS atmospheric CO 2 measurement network for estimating the biogenic CO 2 budget of Europe. Atmospheric Chemistry and Physics, 15(22), 12765–12787. https://doi.org/10.5194/acp-15-12765-2015